# Phylogenomics and metabolic engineering reveal a conserved gene cluster in Solanaceae plants for withanolide biosynthesis

Samuel Edward Hakim [1,2,8], Nancy Choudhary [3,4,8], Karan Malhotra [1,8], Jian Peng[1,2,8], Arne Bültemeier [1,2,8], Ahmed Arafa [1,2,5], Ronja Friedhoff [3], Maximilian Bauer [6], Jessica Eikenberg[1,2], Claus-Peter Witte [7], Marco Herde [7], Philipp Heretsch [6], Boas Pucker [3,4] ✉ & Jakob Franke [1,2] ✉

Withanolides are steroidal lactones from nightshade (Solanaceae) plants with untapped drug potential due to limited availability of minor representatives caused by lack of biosynthetic pathway knowledge. Here, we combine phylogenomics with metabolic engineering to overcome this limitation. By sequencing the genome of the medicinal plant ashwagandha (*Withania somnifera*) and comparing it with nine Solanaceae species, we discover a conserved withanolide biosynthesis gene cluster, consisting of two sub gene clusters with differing expression patterns. We establish metabolic engineering platforms in yeast (*Saccharomyces cerevisiae*) and the model plant *Nicotiana benthamiana* to reconstitute the first five oxidations of withanolide biosynthesis, catalysed by the cytochrome P450 monooxygenases CYP87G1, CYP88C7, and CYP749B2 and a short-chain dehydrogenase/reductase, producing the aglycone of withanoside V. Enzyme functions are conserved within both sub gene clusters in *W. somnifera* and between *W. somnifera* and *Physalis pruinosa*. Our work sets the basis for biotechnological withanolide production to unlock their pharmaceutical potential.

Plants are well-known for their extensive capabilities to produce structurally complex metabolites with potent biological activities. Even though many of these specialised metabolites have been studied extensively at a chemical level, the genetic basis for their biosynthesis is often still unknown, representing a major hurdle for biotechnological improvement of medicinal plants and development of microbial production systems alike. Such a lack of knowledge at the gene level also exists for withanolide biosynthesis. Withanolides are steroidal lactones occurring in several members of the nightshade family (Solanaceae). Named after the first discovery of these compounds in

the medicinal plant *Withania somnifera* (ashwagandha), approx. 1200 withanolides are nowadays known from 22 genera of Solanaceae[1,2]. These metabolites have played a role in traditional medicine since ancient times[3,4]. For example, *W. somnifera* has been used in traditional Indian medicine, Ayurveda, as an anti-stress agent for millennia; these stress-relieving properties are also supported by modern placebo-controlled studies[5–7]. Detailed pharmacological studies underline the broad spectrum of biological activities of withanolides[4,8–12] and withanolide-inspired drug candidates[13]. In stark contrast, only a single enzyme specific to withanolide biosynthesis has been identified and

[1]Institute of Botany, Leibniz University Hannover, 30419 Hannover, Germany. [2]Centre of Biomolecular Drug Research, Leibniz University Hannover, 30167 Hannover, Germany. [3]Institute of Plant Biology & BRICS, TU Braunschweig, 38106 Braunschweig, Germany. [4]Institute for Cellular and Molecular Botany (IZMB), University of Bonn, Kirschallee 1, 53115 Bonn, Germany. [5]Pharmacognosy Department, Faculty of Pharmacy, Tanta University, 31527 Tanta, Egypt. [6]Institute of Organic Chemistry, Leibniz University Hannover, 30167 Hannover, Germany. [7]Department of Molecular Nutrition and Biochemistry of Plants, Leibniz University Hannover, 30419 Hannover, Germany. [8]These authors contributed equally: Samuel Edward Hakim, Nancy Choudhary, Karan Malhotra, Jian Peng, Arne Bültemeier. ✉e-mail: pucker@uni-bonn.de; jakob.franke@botanik.uni-hannover.de

characterised so far[14], hampering efforts to engineer withanolide metabolism *in planta* and produce withanolides biotechnologically[15]. This enzyme, sterol Δ24-isomerase (24ISO)[14], catalyses the isomerisation of 24-methylenecholesterol (**1**), an intermediate of the general phytosterol pathway, to the withanolide-specific key intermediate 24-methyldesmosterol (**2**) (Fig. 1). As such, the 24ISO reaction represents the key committed step at which withanolide biosynthesis branches from phytosterol and brassinosteroid biosynthesis[14]. Several genes and enzymes in the general phytosterol pathway upstream of 24ISO were investigated in withanolide-producing plants[16–19]. In contrast, the biosynthetic pathway downstream of 24ISO, responsible for the conversion of 24-methyldesmosterol (**2**) to withanolides, is not yet known. Several putative withanolide biosynthesis gene candidates were tested by virus-induced gene silencing[20,21], but no clear biochemical activity has been reported for them so far.

While the elucidation of plant biosynthetic pathways has been a slow and tedious process for a long time, advances in sequencing techniques over the past 20 years have dramatically accelerated the pace of gene function discovery in plants[22]. Most commonly, transcriptomic data is used to identify biosynthetic genes based on specific expression patterns in different growth stages, tissues or even cell types[23–26]. Genomic data was traditionally considered to be less important for pathway elucidation, because – in contrast to microorganisms[27] – many biosynthetic genes in plants discovered early were not physically clustered[28–30]. However, this perspective is getting continuously challenged, as more plant genome sequences become available[31]. Indeed, many examples now show that biosynthetic gene clusters in plants are relatively common and enable efficient pathway elucidation[30], as demonstrated recently for saponin, alkaloid, and terpenoid biosynthesis[32–35].

During the discovery of the only withanolide-specific pathway gene, *24ISO*, Knoch et al. reported possible clustering of this gene with other genes common for specialised metabolism[14], but their analysis was still strongly limited by the lack of high-quality genome sequences of Solanaceae plants in 2018. A comparison of *Capsicum annuum*, *Solanum melongena*, and *Petunia inflata* indicated the co-occurrence of *24ISO* with genes encoding cytochrome P450 monooxygenases and oxoglutarate-dependent dioxygenases[14]. However, none of the species of this comparison is known as a producer of canonical withanolides. In recent years, several high-quality genome sequences of withanolide-producing plants have been released. These include for example *Physalis floridana* (*Physalis pubescens*)[36], *Physalis grisea*[37], *Physalis pruinosa*[37], *Datura stramonium*[34,38], and *Datura wrightii*[39].

In this work, we revisit the previous hypothesis of possible gene clustering in the context of withanolide biosynthesis. By sequencing the genome of the archetypical withanolide producer *Withania somnifera* and synteny analyses with other Solanaceae genome sequences, we reveal a conserved gene cluster in withanolide-producing plants that harbours the withanolide pathway gene *24ISO* and multiple other genes typical for specialised metabolism. To overcome previous obstacles in functional validation of withanolide pathway genes, we employ metabolic engineering in the model organisms yeast (*Saccharomyces cerevisiae*) and *Nicotiana benthamiana* to successfully establish two independent platforms for withanolide pathway reconstitution. With these, we characterise three cytochrome P450 monooxygenases and a short-chain dehydrogenase/reductase that oxidise 24-methyldesmosterol (**2**) to construct the pivotal δ-lactone ring of withanolides. Our discovery of a conserved gene cluster for withanolide biosynthesis in Solanaceae plants and the development of synthetic biology systems will enable full elucidation and engineering of withanolide biosynthesis in the future, to further harness the drug potential of withanolides.

## Results

### Genome assembly of *Withania somnifera*

To explore the genomic context of withanolide biosynthesis, we generated a genome assembly of *Withania somnifera*, known as a prolific producer of withanolides. The genome size of *W. somnifera* was estimated to be 2.94 Gb. Building on Oxford Nanopore Technologies' sequencing method, a total of 5.1 million reads (N50: 39.4 kb) corresponding to an estimated genome coverage of 34.6x were sequenced[40]. The de novo assembly comprised 93 contigs with an N50 length of 71 Mb and a total assembly size of 2.88 Gbp (Supplementary Table 1, Supplementary Table 2). The read coverage depth histogram shows a single peak around the estimated genome coverage suggesting a diploid genome with low heterozygosity (Supplementary Fig. 1). A total of 34,955 protein-encoding genes were predicted based on homology and transcriptome data, with an average gene length of 4978 bp and an average coding sequence (CDS) length of 1266 bp. BUSCO analysis[41] using solanales_odb12 dataset revealed 96.2% complete homologues in the predicted proteins of *W. somnifera*. Additionally, we re-annotated the protein-coding genes of the chromosome-scale genome sequences of *P. grisea* and *P. pruinosa*, resulting in improved gene models and higher BUSCO completeness compared to the previous annotations[37]. The predicted genes of both genome sequences achieved 97.5% completeness in the BUSCO analysis.

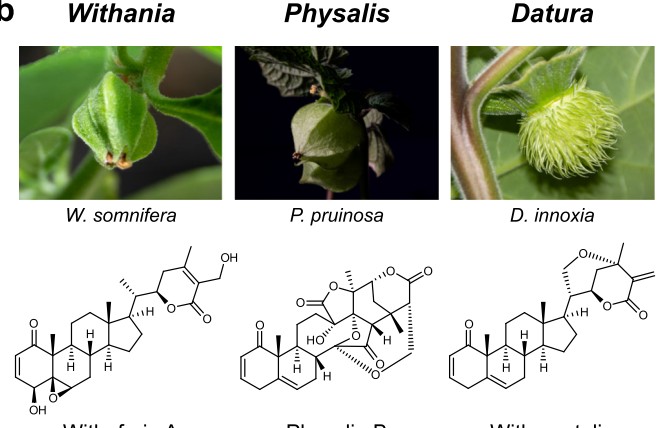

**Fig. 1 | Withanolides are steroidal lactones from Solanaceae plants derived from the general phytosterol pathway intermediate 24-methylenecholesterol (1). a** 24-Methylenecholesterol (**1**) is converted by the only known withanolide biosynthetic enzyme sterol Δ24-isomerase (24ISO)[14] into 24-methyldesmosterol (**2**). The enzymes for all subsequent biosynthetic steps are not known. **b** Representative genera and species of withanolide-producing plants and characteristic compounds from them. Photographs by Jakob Maximilian Horz, TU Braunschweig, Germany.

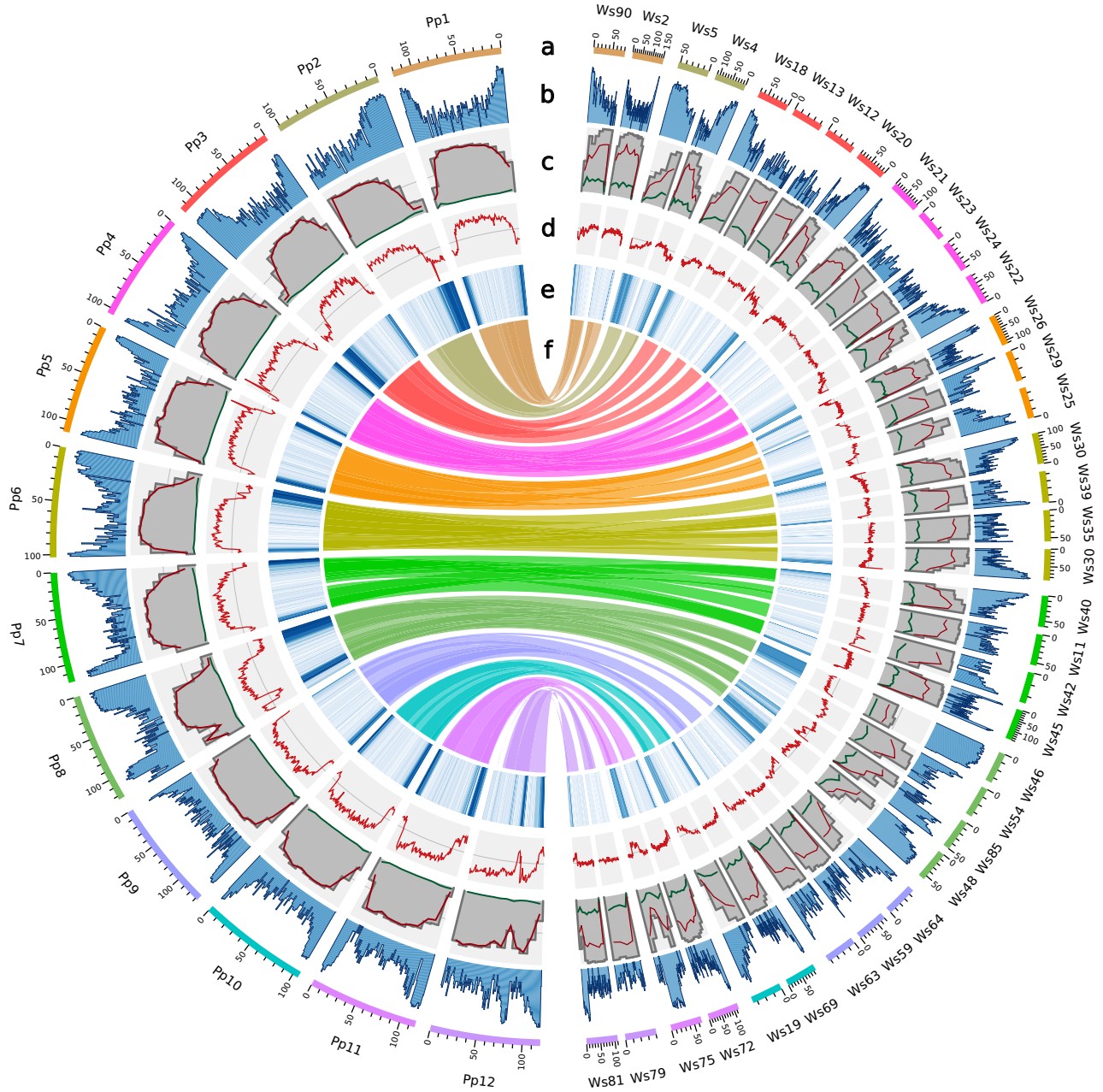

**Fig. 2 | Circos plot comparing important genomic features of the newly assembled genome sequence of *Withania somnifera* (*Ws*, right) with that of *Physalis pruinosa* (*Pp*, left). a** Genomic landscape of the 36 *W. somnifera* pseudochromosomes (right) and 12 *P. pruinosa* pseudochromosomes (left). All density information is calculated in non-overlapping 1-Mbp windows. **b** Tandem repeats density. **c** Percentage of transposable elements (TEs), calculated in 10-Mbp non-overlapping windows. Total TEs in grey, long-terminal repeat (LTR) retrotransposons in red and terminal inverted repeats (TIR) in blue. **d** GC content. **e** Distribution of protein-coding genes. **f** Links between syntenic regions of both genomes.

The 12 pseudochromosomes from the chromosome-scale assembly of *P. pruinosa*[37] were well represented by the top 36 largest contigs in our *W. somnifera* assembly (accounting for ~87% of the total assembly). A Circos plot revealed large overall synteny between both species (Fig. 2). Although the relative amount of repeats is comparable in *P. pruinosa* (77.59%) and *W. somnifera* (76.02%), *W. somnifera* has more terminal inverted repeats (TIR) (13.13% vs. 1.7%).

## Phylogenomics discovery of a putative withanolide biosynthetic gene cluster

With our high-quality assembly of the *W. somnifera* genome sequence in hands, we next set out to systematically explore the genomic context surrounding *24ISO*, the only previously reported withanolide biosynthesis gene. To search for a possible conserved gene cluster, we analysed additional highly continuous genome sequences of the following withanolide-producing plants: *Physalis floridana*[36], *Physalis grisea*[37], *Physalis pruinosa*[37], *Datura stramonium*[34,38], and *Datura wrightii*[39]. For comparison, we included genome sequences of non-withanolide producing Solanaceae species, namely *Solanum lycopersicum*[42] and *Solanum tuberosum*[43] from the same subfamily Solanoideae as well as *Nicotiana tabacum*[44]. Using the experimentally characterised *24ISO* genes[14] as a bait, we identified their genomic positions and further orthologous genes as a starting point for synteny comparison. All genome sequences of withanolide-producing plants analysed here contained two copies of *24ISO* in close proximity. Only in *W. somnifera*, a third *24ISO* copy was additionally found on a different

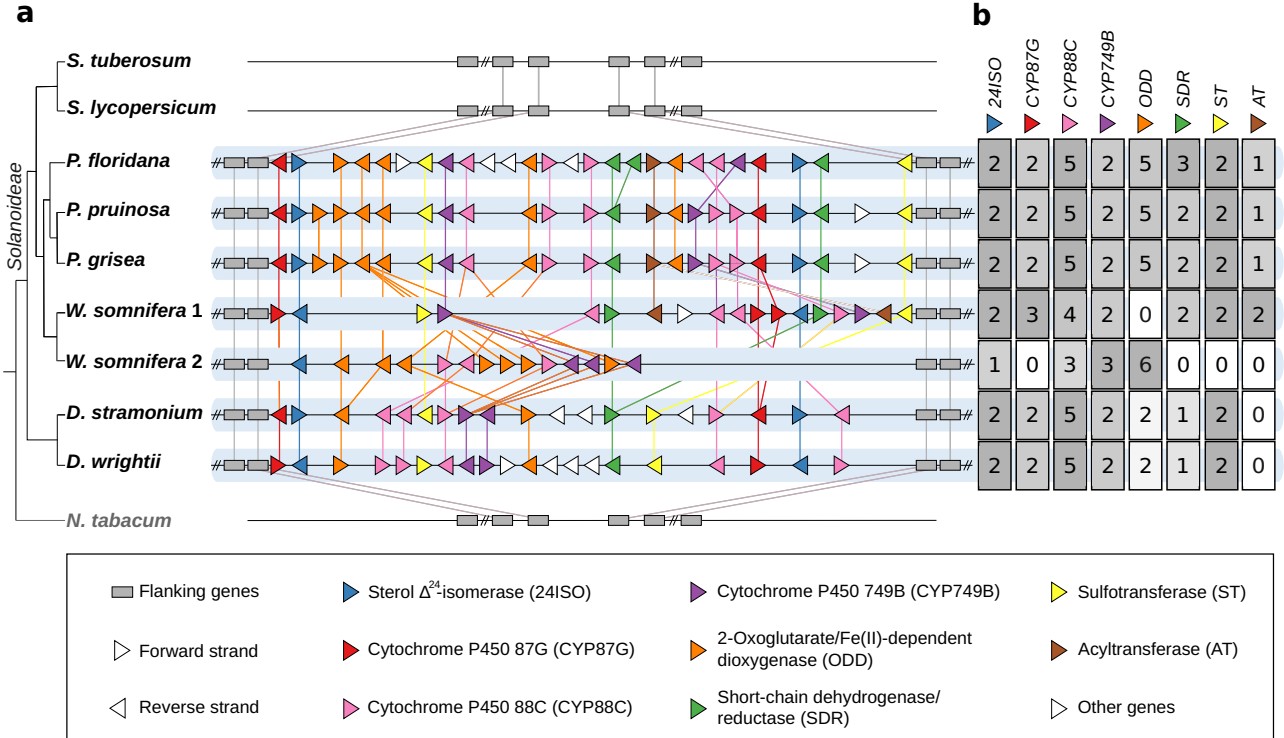

**Fig. 3 | Syntenic biosynthetic gene clusters containing *24ISO* in withanolide-producing Solanaceae plants. a** Synteny plot. Gene lengths are unified for clarity. *N. tabacum* is included as an outgroup of the subfamily Solanoideae. *W. somnifera* 1 and *W. somnifera* 2 refer to the two gene clusters in *W. somnifera* on ctg090 and ctg003, respectively. The phylogenetic relationships shown at the left are adapted from ref. 131. **b** Heatmap summary of gene copy numbers of each gene family. Background colour is normalised to the highest number per column. Expression information about genes in the cluster are displayed in Supplementary Figs. 9–11.

contig. No *24ISO* orthologue was found in *S. lycopersicum*, *S. tuberosum*, and *N. tabacum*. Then, we compared the synteny of the genomic regions surrounding these *24ISO* orthologues (Fig. 3, Supplementary Fig. 2, Supplementary Data 1). Genomes of withanolide producers contained a syntenic region that was absent in the non-producers *S. lycopersicum* and *S. tuberosum*. We then deduced the classes of encoded enzymes in this syntenic region by their Pfam domains. Strikingly, all genes in this syntenic region belong to gene families common in plant specialised metabolism, most importantly cytochrome P450 monooxygenases (CYPs), 2-oxoglutarate-dependent dioxygenases (ODDs), short-chain dehydrogenases/reductases (SDRs), and acyltransferases (AT). CYPs are particularly well-known for their central role in triterpenoid and steroid biosynthetic pathways[26,45–47]. The CYPs in the putative withanolide biosynthetic gene cluster fall into three different CYP families, namely CYP87, CYP88 (both part of the CYP85 clan) and CYP749 (part of the CYP72 clan). Both clans, particularly CYP85, are well-known as hotspots of CYPs involved in triterpenoid and steroid metabolism[45]. Less expected was the occurrence of sulfotransferase (ST) genes. While sulfotransferases are well-known in the biosynthesis of certain specialised metabolites such as glucosinolates[48], a possible link to withanolide biosynthesis is not yet known. A few withanolides bearing 3-*O*-sulphate groups are known, however[49,50], and their biosynthesis might involve a dedicated sulfotransferase. Phylogenetic analyses of the identified genes revealed that these genes from withanolide-producing species clustered together, forming a distinct clade; no orthologues are found in non-withanolide producing species like tomato, potato, and *Nicotiana* (Supplementary Figs. 3–8).

Next, we analysed published RNA-seq datasets to verify that the genes in the gene cluster are not only physically clustered, but also co-expressed. Surprisingly, we observed two groups with distinct expression patterns, both in *W. somnifera* and in *D. stramonium*

(Supplementary Figs. 9–11). This suggests that the withanolide gene cluster is separated into two sub gene clusters which are differentially regulated. Taken together, our phylogenomics analysis suggested that withanolide biosynthesis involves a conserved gene cluster.

## Development of a yeast platform for production of withanolide pathway intermediates

Next, we wanted to support our discovery of a putative withanolide biosynthetic gene cluster by elucidating the biochemical functions of pathway enzymes. Reconstitution of withanolide biosynthesis in heterologous hosts has remained an unsolved problem up until now. The very low polarity and the limited accessibility of the last known intermediate 24-methyldesmosterol (**2**) prevents an efficient use as a substrate for enzyme assays in vitro. An alternative would be to produce 24-methyldesmosterol (**2**) in vivo, but no efficient metabolic engineering strategy has been reported so far. Previous studies showed that its precursor, 24-methylenecholesterol (**1**), can be produced in yeast (*Saccharomyces cerevisiae*) by deleting the ergosterol biosynthesis genes *ERG4* and *ERG5* and adding a gene encoding a Δ7 reductase from plants or animals, in order to hijack the sterol metabolism of yeast to produce plant-like sterols[51–53]. We decided to utilise and expand this strategy to set up a platform for functional evaluation of withanolide biosynthesis genes in yeast (Fig. 4a).

With the help of established CRISPR/Cas techniques[54], we inserted *Physalis peruviana* orthologues of the known plant genes *sterol Δ7 reductase* (*7RED*, also known as *DWF5*)[55] and *24ISO*[14] into the genome of the prototrophic *S. cerevisiae* strain ST7574, which is derived from CEN.PK113-7D and contains a *cas9* gene[54]. The resulting strain KMY14 only produced trace amounts of 24-methylenecholesterol (**1**) and 24-methyldesmosterol (**2**) (Fig. 4b). This result was in line with previous reports showing that under native conditions the ergosterol biosynthetic enzymes ERG4 and ERG5 efficiently consume the

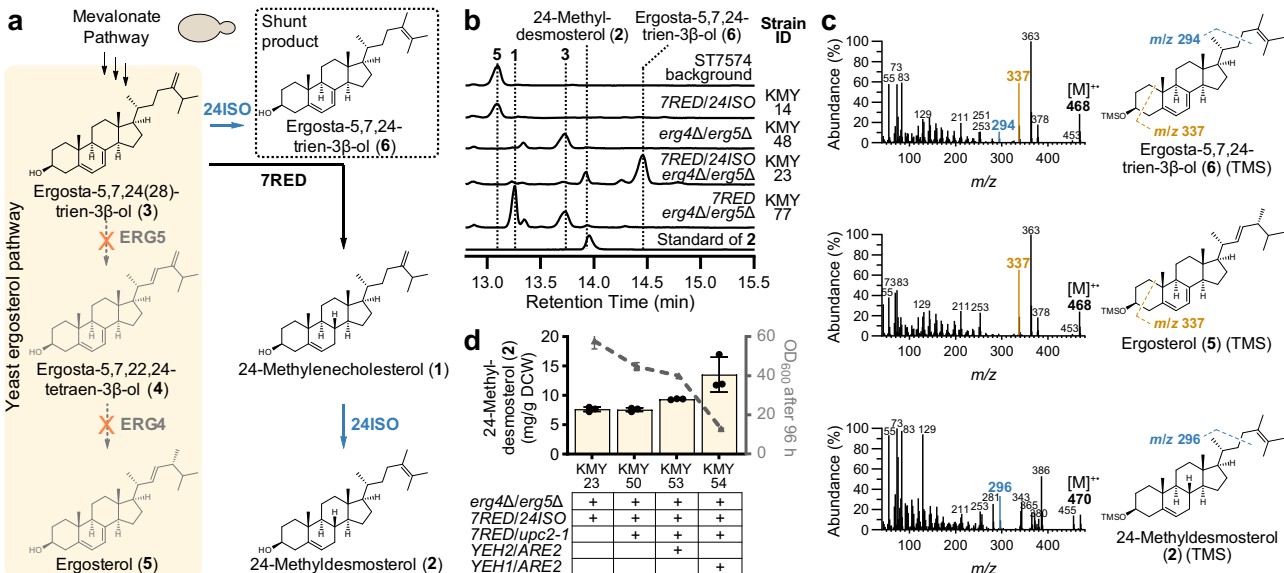

**Fig. 4 | Metabolic engineering of yeast for producing the key intermediate 24-methyldesmosterol (2). a** Yeast engineering strategy to divert flux from ergosterol (**5**) biosynthesis to 24-methyldesmosterol (**2**). **b** GC-MS total ion current chromatograms showing production of 24-methyldesmosterol (**2**) and the shunt product ergosta-5,7,24-trien-3β-ol (**6**) in engineered yeast strain KMY23. **c** Comparison of electron impact mass spectra of ergosta-5,7,24-trien-3β-ol (**6**) and related compounds as TMS ethers supporting the proposed structure of **6**. **d** Further yeast engineering to improve 24-methyldesmosterol (**2**) production. The bar plot shows means ± SD and data points of three independently grown yeast cultures. DCW: Dry cell weight. Budding_yeast icon by umasstr (https://github.com/umasstr) is licensed under CC0 (https://creativecommons.org/publicdomain/zero/1.0/) and was used without modifications. Source data are provided as a Source Data file.

shared intermediate ergosta-5,7,24(28)-trien-3β-ol (**3**) via ergosta-5,7,22,24(28)-tetraen-3β-ol (**4**) to ergosterol (**5**)[14,52,53]. Upon deletion of *ERG4* and *ERG5* in the background strain ST7574 (strain KMY48), the required shared intermediate ergosta-5,7,24(28)-trien-3β-ol (**3**) was formed as the major sterol as reported in literature[56,57]. To redirect this intermediate to the desired product 24-methyldesmosterol (**2**), we deleted *ERG4* and *ERG5* in the *7RED/24ISO*-expressing strain KMY14 to generate KMY23. Gratifyingly, KMY23 produced 24-methyldesmosterol (**2**) at a level of 7.6 mg/g dry cell weight (DCW) (Fig. 4b). Besides 24-methyldesmosterol (**2**), a major peak **6** at 14.5 min was observed in yeast, with a mass difference of −2 compared with 24-methyldesmosterol (**2**) (Fig. 4b). The mass spectrum of this compound **6** was almost identical to that of ergosterol (**5**) (Fig. 4c), but the difference in retention time indicated that **6** must be an isomer of ergosterol (**5**) (Fig. 4b). A striking difference in their mass spectra was the presence of a fragment at *m/z* 294 for **6** that was absent for ergosterol (**5**); in the mass spectrum of 24-methyldesmosterol (**2**), this fragment was shifted to *m/z* 296, matching the mass difference of the molecular ions. Previous studies reported that this fragment is indicative of sterols with a Δ24(28) or Δ24(25) double bond and is formed by a McLafferty rearrangement involving allylic cleavage of the C-22(23) bond[57–60]. The mass difference of −2 suggested that **6** must possess an additional double bond in the ABCD ring system. Ergosterol (**5**) and **6** both exhibit a major fragment at *m/z* 337 ([M − 131]⁺), which is characteristic of sterols with Δ5,7 dienes[61,62]; therefore, the additional double bond of **6** is most likely positioned at C-7,8. We therefore propose that compound **6** is ergosta-5,7,24-trien-3β-ol (Fig. 4c). This Δ7 analogue of 24-methyldesmosterol (**2**) would be formed if ergosta-5,7,24(28)-trien-3β-ol (**3**) is converted by 24ISO before reduction by 7RED can take place. This hypothesis was also supported by the fact that upon deletion of *24ISO* the peak for ergosta-5,7,24-trien-3β-ol (**6**) completely disappeared (strain KMY77); instead, a mixture of the 7RED product 24-methylenecholesterol (**1**) and non-reduced ergosta-5,7,24(28)-trien-3β-ol (**3**) was observed (Fig. 4b). The occurrence of the non-reduced but isomerised product ergosta-5,7,24-trien-3β-ol (**6**) in KMY23 and non-reduced product ergosta-5,7,24(28)-trien-3β-ol (**3**) in KMY77 therefore indicates a misbalance between the very high activity of

*P. peruviana* 24ISO and the limited activity of *P. peruviana* 7RED in our yeast system.

To facilitate subsequent gene discovery, we next wanted to improve the production of 24-methyldesmosterol (**2**) in yeast. In initial experiments, we observed that overexpression of mevalonate pathway genes only resulted in increased levels of squalene but not of downstream sterols. Therefore, to keep the metabolic burden on our strains as low as possible, no mevalonate pathway genes were overexpressed. Instead, we first added an additional copy of *P. peruviana 7RED* to improve the conversion of ergosta-5,7,24(28)-trien-3β-ol (**3**) to 24-methylenecholesterol (**1**); simultaneously, we overexpressed the *upc2-1* allele, encoding transcription factor mutant UPC2^G888D involved in the regulation of sterol metabolism and reported to boost the biosynthesis of sterols[63,64]. The resulting strain KMY50, however, did not show elevated levels of 24-methyldesmosterol (**2**). Previous studied showed that manipulation of sterol homeostasis, i.e., the balance of sterol acylation and sterol ester hydrolysis, can improve the production of sterols in yeast[65]. We therefore co-expressed the acyl-CoA:sterol acyltransferase gene *ARE2* either with the sterol ester hydrolase gene *YEH2* (KMY53) or with *YEH1* (KMY54). In both cases, an improvement in 24-methyldesmosterol (**2**) levels was noted. KMY53 produced 9.3 mg/g DCW, whereas KMY54 reached 13.5 mg/g DCW (Fig. 4d). However, the final OD600 after cultivation for 96 h in shake flasks was severely reduced, particularly for strain KMY54. In summary, our data show that the key intermediate 24-methyldesmosterol (**2**) can be produced in engineered yeast, but further strain improvement is required to improve product levels, reduce the amounts of shunt product **6**, and overcome growth defects.

### Engineering of *Nicotiana benthamiana* for production of 24-methyldesmosterol

Due to these unresolved limitations of yeast as a platform for withanolide pathway reconstitution, we alternatively envisioned to generate a plant-based system using the popular model organism *Nicotiana benthamiana*[66]. In contrast to yeast, plants natively produce 24-methylenecholesterol (**1**) as a transient intermediate en route to phytosterols and brassinosteroids[67]. We therefore expected that usage

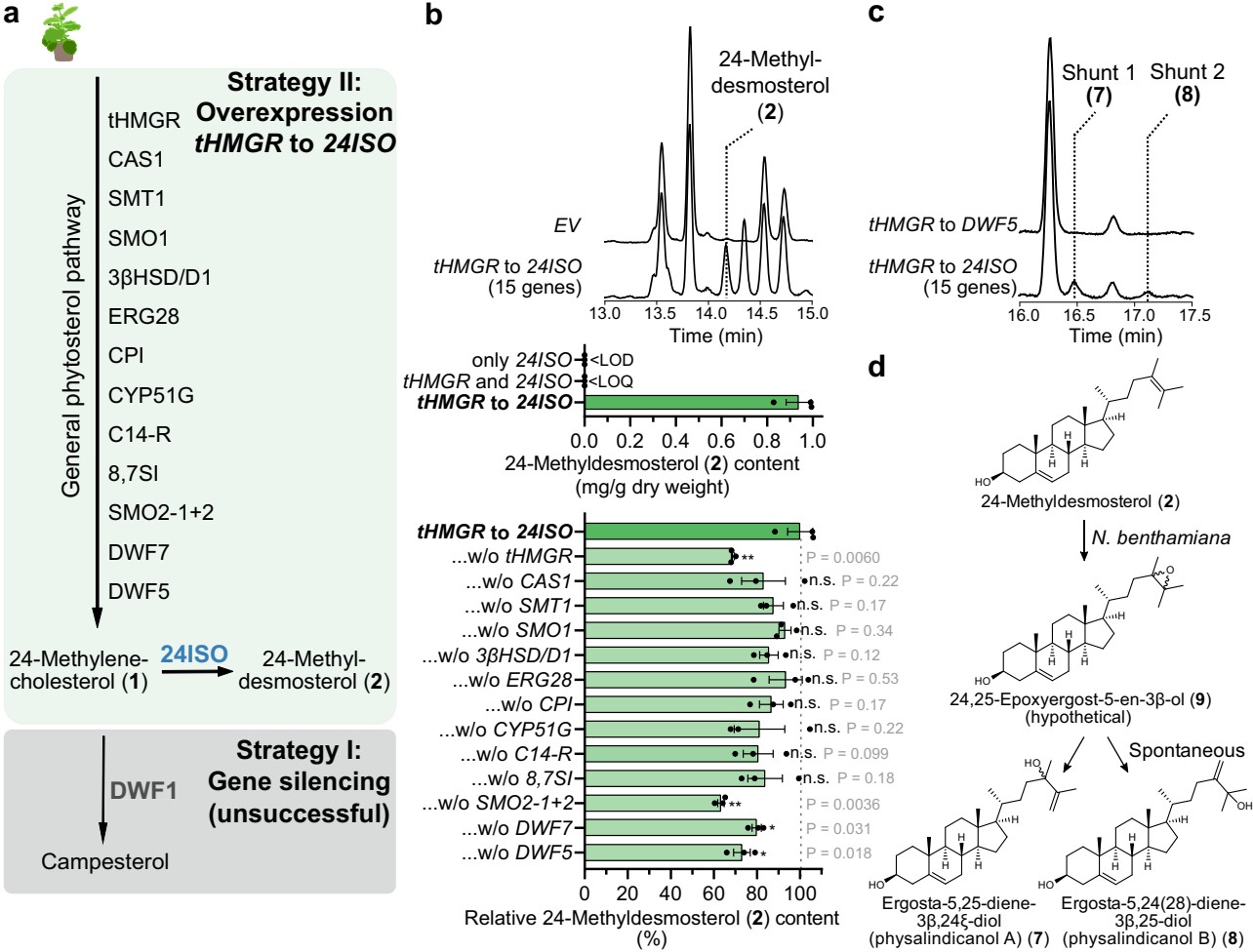

**Fig. 5 | Metabolic engineering in *Nicotiana benthamiana* for producing the key intermediate 24-methyldesmosterol (2). a** Overview of the two different engineering strategies that were attempted in this work. **b** Effect of phytosterol pathway gene overexpression (strategy II) on production of 24-methyldesmosterol (**2**). Total ion current (TIC) GC-MS chromatograms of representative samples from transient expression in *N. benthamiana*. The bottom bar plot indicates relative 24-methyldesmosterol (**2**) levels when a single gene of the 15-gene set from *tHMGR* to *24ISO* was left out. Bar plots show means ± SEM and data points of three biological replicates. * *P* < 0.05, ** *P* < 0.01, NS not significant by Student's two-tailed unpaired

*t* test; exact P values are indicated. LOD: limit of detection, LOQ, limit of quantification. **c** GC-MS TIC chromatograms showing formation of two shunt products **7** and **8** dependent on *24ISO* in *N. benthamiana*. **d** Proposed formation of shunt products **7** and **8** by background epoxidation of the Δ²⁴ double bond in *N. benthamiana*. Nicotiana_benthamiana icon by Connor-Tansley (https://github.com/Ctansley) is licensed under CC-BY 4.0 Unported (https://creativecommons.org/licenses/by/4.0/) and was used without modifications. Source data are provided as a Source Data file.

of a plant host would enable us to circumvent the undesired yeast shunt product ergosta-5,7,24-trien-3β-ol (**6**) from the misbalance of 24ISO and 7RED activity. Furthermore, transient expression in *N. benthamiana* provides the added benefit of easy and flexible "mix-and-match" co-expression, which would drastically accelerate the pace of testing different gene combinations[66]. We first transiently expressed only *24ISO*, the gene encoding sterol Δ²⁴-isomerase[14], in *N. benthamiana*; while this caused changes in the metabolic profile, no 24-methyldesmosterol (**2**) was detected. This suggested that insufficient amounts of 24-methylenecholesterol (**1**) accumulate under native conditions in *N. benthamiana*. We therefore sought to increase the formation of 24-methylenecholesterol (**1**) and hence 24-methyldesmosterol (**2**) by metabolic engineering. In principle, this could be achieved by two complementary strategies: Either by blocking the pathway downstream of 24-methylenecholesterol (**1**) controlled by the reductase DWF1[68] (strategy I), or overexpressing its upstream phytosterol pathway (strategy II) (Fig. 5a). We first attempted to silence *DWF1* by virus-induced gene silencing[69]. In accordance with previous reports[68], this caused a very strong dwarf phenotype and sick-

looking plants (Supplementary Fig. 12a), probably due to the effects on brassinosteroid levels. After transient overexpression of *24ISO* in *DWF1*-silenced *N. benthamiana* plants, small quantities of 24-methyldesmosterol (**2**) could be detected (Supplementary Fig. 12b). Nonetheless, the experimental challenges of coordinating the relatively slow systemic gene silencing with the relatively fast local transient overexpression and the severe phenotypic effects on the plants prompted us to test upstream pathway overexpression as an alternative strategy (Fig. 5a). First, we co-expressed *24ISO* with a gene encoding truncated, feedback-insensitive hydroxymethylglutaryl coenzyme A reductase (tHMGR), which is known as a very efficient booster for the mevalonate pathway and triterpenoid production[46]. In these samples, we observed a very small peak below the limit of quantification corresponding to 24-methyldesmosterol (**2**) (Fig. 5b). We concluded that further reactions of the phytosterol pathway limit the 24-methyldesmosterol (**2**) yield.

As no comprehensive information is available which enzymes en route to 24-methylenecholesterol (**1**) are rate-limiting, we decided to transiently overexpress the full gene set for the phytosterol pathway

reported from *Arabidopsis thaliana* (Supplementary Fig. 13) in addition to *tHMGR* and *24ISO* in *N. benthamiana*. This comprised a total of 15 genes. Gratifyingly, this approach led to a drastic increase in 24-methyldesmosterol (**2**) formation to 0.9 mg/g dry weight (Fig. 5b). Notably, the byproduct ergosta-5,7,24-trien-3β-ol (**6**) was not observed, supporting our hypothesis that the 24ISO/7RED misbalance issue encountered in yeast can be circumvented by using a plant host. To understand which upstream pathway genes had the largest effect on 24-methyldesmosterol (**2**) titres, we performed a leave-one-out experiment (Fig. 5b). Statistically significant negative effects on 24-methyldesmosterol (**2**) levels were observed when *tHMGR, SMO2-1* plus *SMO2-2, DWARF5,* or *DWARF7* were not included, suggesting that these represent major bottlenecks towards 24-methylenecholesterol (**1**).

The chromatograms of *N. benthamiana* plants producing 24-methyldesmosterol (**2**) also contained two new product peaks dependent on the presence of *24ISO* with a mass difference of +88 compared with 24-methyldesmosterol (**2**), which would fit to an extra trimethylsilyl (TMS)-bearing oxygen (Fig. 5c). Analysis of fragmentation patterns suggested that these compounds were oxidised in the side chain (Supplementary Fig. 14-16, Supplementary Data 2). The two compounds **7** and **8** were successfully isolated (isolated yield 0.03 and 0.03 mg/g dry weight, respectively) and fully characterised by NMR spectroscopy (Supplementary Fig. 17, Supplementary Data 3). Surprisingly, in comparison to 24-methyldesmosterol (**2**), the Δ24 double bond of **7** and **8** was shifted to Δ25 or Δ24(28), respectively. In agreement with the fragmentation analysis, NMR data showed that both products **7** and **8** contained an additional hydroxy group in allylic position of the double bond in the side chain. Hence, the systematic name of **7** is ergosta-5,25-diene-3β,24ξ-diol, while **8** is ergosta-5,24(28)-diene-3β,25-diol. Both compounds were isolated before from the withanolide-producing plant *Physalis minima* var. *indica* and named physalindicanol A (**7**) and B (**8**), respectively[70]. The occurrence of these regioisomeric allylic alcohols can be explained by background epoxidation of the Δ24 double bond of 24-methyldesmosterol (**2**)[70] in *N. benthamiana*, generating hypothetical intermediate **9**; either in vivo or during workup, this epoxide can then be opened to allylic alcohols **7** and **8** (Fig. 5d).

We concluded that, despite the undesirable background epoxidation, our metabolic engineering strategy in *N. benthamiana* reaching 0.9 mg/g dry weight of 24-methyldesmosterol (**2**) was suitable for further elucidation of withanolide biosynthesis.

## Dihydroxylation of 24-methyldesmosterol in withanolide biosynthesis

After establishing heterologous platforms capable of producing the last known withanolide biosynthesis intermediate 24-methyldesmosterol (**2**), we turned our attention to screening genes from the gene cluster (Fig. 3) to identify the next steps in withanolide biosynthesis. Labelling studies demonstrated that withanolide biosynthesis proceeds by oxidative assembly of the side chain lactone[71,72]. Genes from the cytochrome P450 monooxygenase subfamilies CYP87G, CYP88C, and CYP749B were well conserved within all withanolide-producing species (Fig. 3, Supplementary Fig. 18) and therefore prioritised. To test if CYPs showed functional conservation between sub gene clusters and also between species, we selected CYP homologues not only from our main model system *W. somnifera* (*Ws*) but also from *P. pruinosa* (*Pp*) for a first rapid screening in our *N. benthamiana* platform producing 24-methyldesmosterol (**2**).

Of the tested CYP genes, only those from the *CYP87G* subfamily showed activity on 24-methyldesmosterol (**2**) detectable by GC-MS. The same activity was observed for the two *W. somnifera* homologues from the two sub gene clusters and for the tested homologue from *P. pruinosa* (Supplementary Fig. 19a). As all of these functionally equivalent CYPs belong to the same orthogroup, they were assigned the same systematic name CYP87G1 and were further distinguished by species codes (*Ws* or *Pp*) and internal names (see Supporting File). All

subsequent experiments were carried out with the *CYP87G1* homologue from *P. pruinosa* (*PpCYP87G1*). To test the function of *Pp*CYP87G1 in yeast, the 24-methyldesmosterol (**2**) producing strain KMY23 in combination with a cytochrome P450 reductase gene from *Arabidopsis thaliana* was used (strain KMY55).

In both *N. benthamiana* and yeast, a new product peak **10** with a mass shift of +88 was detected by GC-MS upon presence of *Pp*CYP87G1 (Fig. 6b). Analysis of the mass spectrum from electron impact ionisation suggested that **10** might be a hydroxylation product of 24-methyldesmosterol (**2**) at C-22 (Supplementary Fig. 14-16, Supplementary Data 2). We isolated the shared product **10** from *N. benthamiana* (0.04 mg/g dry weight isolated yield) and elucidated its structure by NMR spectroscopy (Supplementary Fig. 17, Supplementary Data 3). In agreement with our GC-MS fragmentation analysis, **10** contained an additional hydroxy group at C-22. Due to the flexibility and free rotation of the side chain, the stereochemistry at C-22 could not be confidently deduced by nuclear Overhauser effect spectroscopy (NOESY). Instead, we compared the coupling pattern of H-22 of our isolated compound **10** with two pairs of structurally related C-22 epimers that were obtained by semi synthesis (Supplementary Fig. 20). For the two synthetic 22*S* epimers, doublets of doublets were observed at H-22, in contrast to apparent doublets of triplets for the two synthetic *22R* epimers. As compound **10** also showed an apparent doublet of triplets for H-22, C-22 was assigned as 22*R*, which is also the configuration expected for natural withanolides. In conclusion, compound **10** was confirmed to be (22*R*)-ergosta-5,24-diene-3β,22-diol (22*R*-hydroxy-24-methyldesmosterol). Besides shared product **10**, we also observed further compounds from *PpCYP87G1* expression occurring exclusively either in yeast or in *N. benthamiana*. In yeast, a second compound **11** with a mass shift of −2 compared with **10** was produced, while in *N. benthamiana* further compounds **12** and **13** with mass differences of +88 and +16 compared with **10** were detected (Fig. 6b). As the exclusive occurrence in either yeast or *N. benthamiana* as well as the mass shifts were in excellent agreement with the occurrence of shunt products related to 24-methyldesmosterol (**2**) observed during our metabolic engineering efforts (Figs. 4b, 5c), we strongly suspected that these products reflected the same background reactions and not additional enzymatic activity of *Pp*CYP87G1. To confirm this, we isolated the putative shunt products **12** and **13** from *N. benthamiana* (0.04 and 0.01 mg/g dry weight isolated yields). NMR spectroscopy indicated that both products contained a hydroxy group at C-22 (Supplementary Fig. 17, Supplementary Data 3). In addition, compound **12** possessed the same Δ24(28) double bond and C-25 hydroxy group as shunt product **8**. Compound **13** did not contain any olefinic carbons in the side chain, but instead two carbons at 62.6 and 65.6 ppm, respectively, which imply that **13** contains an epoxy group. The isolation of epoxide **13** strongly supports our hypothesis that the rearranged allylic alcohols are derived from Δ24 epoxidation (Fig. 5d). Compound **12**, named (22*R*)-ergosta-5,24(28)-diene-3β,22,25-triol was isolated before from *Physalis minima* and has been named phyministerol A[73]; our NMR data of **12** are in very good agreement with the published data (Supplementary Table 3). Compound **13** or (22*R*)-24,25-epoxyergost-5-ene-3β,22-diol has not been reported before, but steroids with matching side chains have been isolated from the nightshade plant *Petunia hybrida* (Supplementary Fig. 21)[74].

Next, we co-expressed *PpCYP87G1* with the remaining CYP gene candidates in *N. benthamiana* to find the next step in the pathway. Only upon co-expression of CYPs of the orthogroup *CYP88C7* we observed new product peaks by GC-MS (Supplementary Fig. 19b). Again, the same activity was observed for both *CYP88C7* homologues from the *W. somnifera* sub gene clusters and for the tested homologue from *P. pruinosa* (Supplementary Fig. 19b). Notably, none of the other two tested *CYP88C* genes belonging to a different orthogroup showed any activity under these conditions (Supplementary Fig. 19b). We also

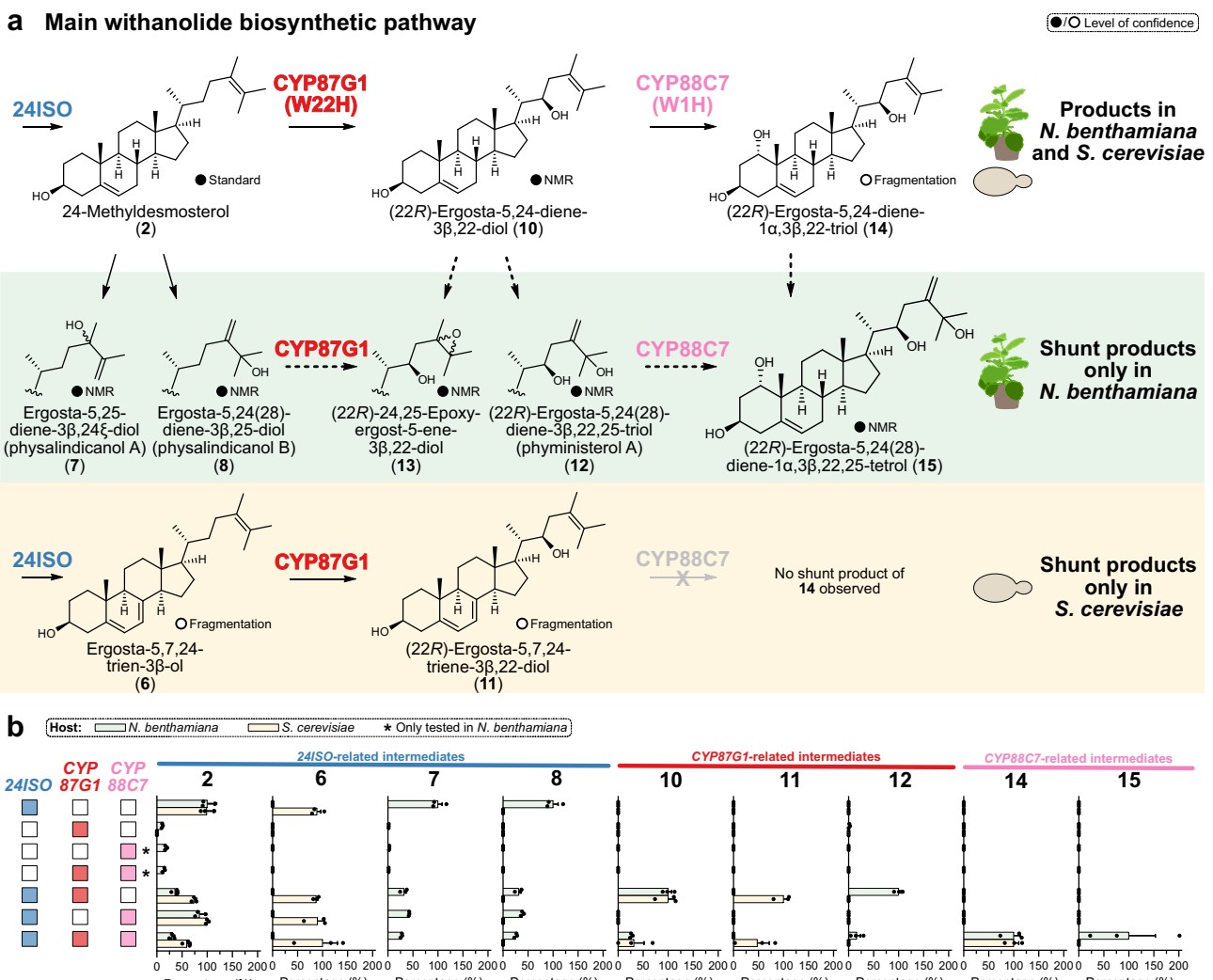

**Fig. 6 | Dihydroxylation of 24-methyldesmosterol (2) by CYP87G1 (W22H) and CYP88C7 (W1H). a** Proposed formation of intermediates and shunt products in *N. benthamiana* and *S. cerevisiae* during the dihydroxylation of 24-methyldesmosterol (**2**). Dashed arrows indicate alternative pathways to *N. benthamiana* shunt products. **b** Relative levels of identified compounds in GC-MS chromatograms of *N. benthamiana* (green) and *S. cerevisiae* (yellow) upon co-expression of genes from *P. pruinosa* (*Pp*). For each compound, peak areas were normalised to internal standard and sample dry weight (either *N. benthamiana* leaf dry weight or yeast dry cell weight) and converted to relative amounts by setting the highest mean value of each compound

for each host organism to 100%. Bar plots show means ± SEM and data points of three biological replicates. W22H: withanolide biosynthesis 22-hydroxylase; W1H: withanolide biosynthesis 1-hydroxylase. Budding yeast icon by umasstr (https://github.com/umasstr) is licensed under CC0 (https://creativecommons.org/publicdomain/zero/1.0/) and was used without modifications. Nicotiana benthamiana icon by Connor-Tansley (https://github.com/Ctansley) is licensed under CC-BY 4.0 Unported (https://creativecommons.org/licenses/by/4.0/) and was used without modifications. Source data are provided as a Source Data file.

introduced *Pp*CYP88C7 into our yeast strain already harbouring *Pp*CYP87G1 for further evaluation of the biochemical activity. One product **14** with a mass shift of +88 by GC-MS analysis compared with the *Pp*CYP87G1 product **10** was found in both heterologous hosts. In *N. benthamiana*, extra peaks with an additional +88 mass shift were observed; no extra peak was detected in yeast. The mass spectra of all *Pp*CYP88C7-dependent peaks showed a *m/z* 217 fragment, which has been reported as a diagnostic ion for 1,3-dihydroxylated cyclohexanes[75] (Supplementary Figs. 14–16, Supplementary Data 2). The fragmentation pattern therefore suggested that CYP88C7 carries out hydroxylation of C-1. Although we could not purify sufficient amounts of **14** for NMR spectroscopy, we successfully obtained shunt product **15** from *N. benthamiana* (isolated yield 0.01 mg/g dry weight). NMR data indicated that the side chain of **15** was identical to shunt product (22*R*)-ergosta-5,24(28)-diene-3β,22,25-triol (phyministerol A) (**12**). However, one of the other carbons exhibited a drastic downfield shift to 73.1 ppm, indicating an additional hydroxy substituent. An

HMBC correlation from methyl group H-19 and COSY correlations with H-2 unambiguously confirmed that the new hydroxy group was located at C-1 (Supplementary Fig. 17). A NOE correlation between methyl group H-19 and H-1 indicated an α configuration of C-1 (Supplementary Fig. 22). Previously isolated steroids from nightshade plants with a hydroxy group at C-1, such as withanosides, also possess a C-1 α configuration[76].

## Elucidation of oxidative lactone formation in withanolide biosynthesis

As the next step in the biosynthetic pathway, we speculated that the intermediate **14** would be oxidatively transformed to the corresponding lactone compound, which would correspond to the aglycone of the known withanolide glycoside withanoside V (**16**)[76] (Fig. 7a). We therefore screened our remaining CYP candidate genes in combination with *Pp*CYP87G1 and *Pp*CYP88C7 in our *N. benthamiana* system. Only when *Pp*CYP87G1, *Pp*CYP88C7, and *Pp*CYP749B2 were co-

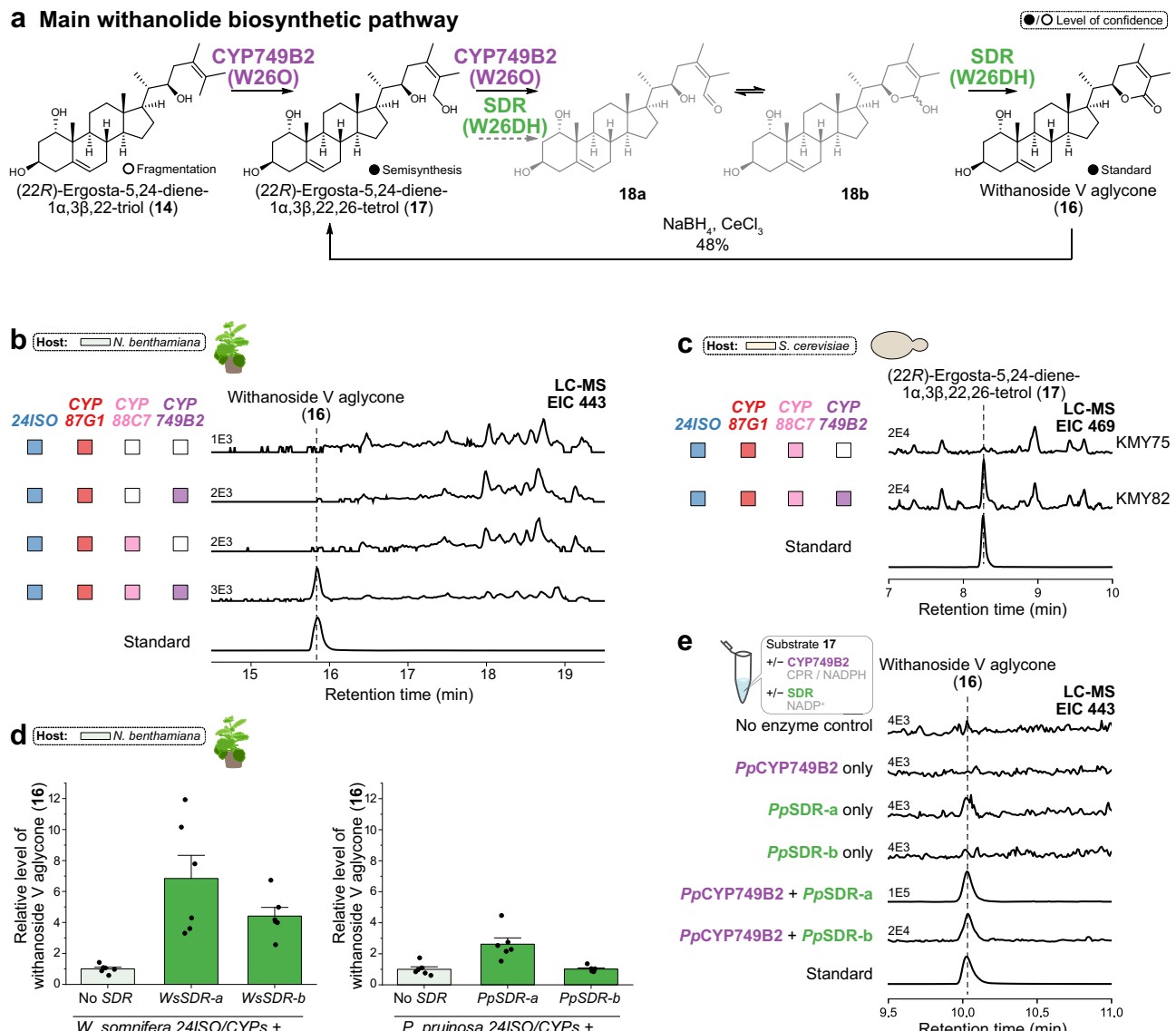

**Fig. 7 | Lactone formation by CYP749B2 (W26O) and SDR (W26DH) generating withanoside V aglycone (16). a** Proposed lactone formation from (22*R*)-ergosta-5,24-diene-1α,3β,22-triol (**14**) to withanoside V aglycone (**16**) by CYP749B2 (W26O) and SDR (W26DH) and semisynthetic conversion of **16** to (22*R*)-ergosta-5,24-diene-1α,3β,22,26-tetrol (**17**). **b** Co-expression of *CYPs* from *P. pruinosa* (*Pp*) in *N. benthamiana* leading to formation of withanoside V aglycone (**16**) in comparison to an authentic reference compound. Extracted ion LC-MS chromatograms of *m/z* 443 (low resolution) are shown with the respective ion levels. **c** Co-expression of *CYPs* from *P. pruinosa* (*Pp*) in yeast leading to formation of (22*R*)-ergosta-5,24-diene-1α,3β,22,26-tetrol (**17**). Extracted ion LC-MS chromatograms of *m/z* 469 are shown with the respective ion levels. Withanoside V aglycone (**16**) was not detected in these strains and was only produced when *SDR* was additionally included (Supplementary Fig. 24). **d** Increased formation of withanoside V aglycone (**16**) in *N. benthamiana* upon co-expression of *SDR* homologues with *24ISO* and *CYP87G1/CYP88C7/CYP749B2* genes. Comparable results were obtained when genes

only from *W. somnifera* (left) or only from *P. pruinosa* (right) were used, with the exception of *PpSDR-b*. The bar plots show means ± SEM and data points of six biological replicates. **e** In vitro assays with *Pp*CYP749B2 (obtained as yeast microsomes containing a cytochrome P450 reductase (CPR)) and *Pp*SDR-a/b (purified soluble protein produced in *E. coli*). Semisynthetic intermediate **17** was used as a substrate. Extracted ion LC-MS chromatograms of *m/z* 443 to monitor formation of withanoside V aglycone (**16**) are shown with the respective ion levels. W26O: Withanolide biosynthesis 26-oxidase; W26DH: withanolide biosynthesis 26-dehydrogenase. Budding_yeast icon by umasstr (https://github.com/umasstr) is licensed under CC0 (https://creativecommons.org/publicdomain/zero/1.0/) and was used without modifications. Nicotiana_benthamiana icon by Connor-Tansley (https://github.com/Ctansley) is licensed under CC-BY 4.0 Unported (https://creativecommons.org/licenses/by/4.0/) and was used without modifications. Source data are provided as a Source Data file.

expressed, a new peak was observed by LC-MS that matched withanoside V aglycone (**16**) in terms of retention time, high-resolution mass and MS/MS fragmentation (Fig. 7b, Supplementary Fig. 23). This compound was not observed when samples were saponified before LC-MS analysis. The function of CYP749B2 from *W. somnifera* and *P. pruinosa* was conserved; surprisingly, though, only one of the two homologues from the *W. somnifera* gene cluster 1 showed activity (Supplementary Fig. 19c). Co-expression of *PpCYP87G1* and

*PpCYP749B2* without *PpCYP88C7* did not result in lactone formation, suggesting that lactone formation in the side chain cannot take place before C-1 hydroxylation.

To further corroborate the role of CYP749B2 for lactone formation, we introduced *PpCYP749B2* into our previous yeast strain already containing *PpCYP87G1* and *PpCYP88C7*. Surprisingly, no withanoside V aglycone (**16**) was detectable in this system. We first suspected that weak CYP activity as often observed in yeast was the reason for this

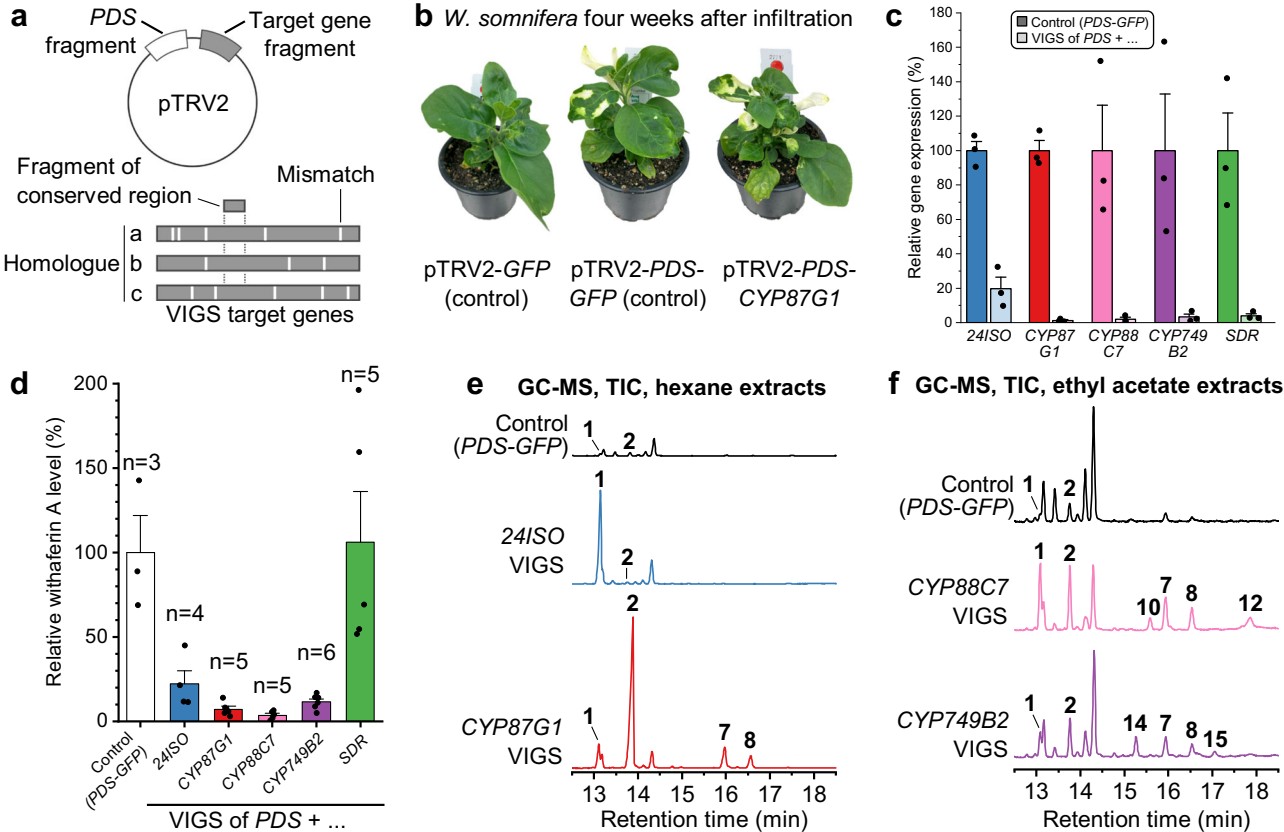

**Fig. 8 | Virus-induced gene silencing in *W. somnifera* confirms the role of the biosynthetic gene cluster for withanolide biosynthesis. a** Construct design to co-silence all homologues of target genes and the visual marker gene phytoene desaturase (*PDS*). **b** Representative pictures of *W. somnifera* plants infiltrated with silencing constructs four weeks after infiltration. **c** Quantitative reverse transcription polymerase chain reaction analysis of target gene expression in silenced leaves. The bar plot shows mean ± SEM and data points of three biological replicates. **d** Relative quantification of withaferin A levels in silenced plants. The bar plot shows mean ± SEM and individual data points; the sample size of replicates showing successful silencing as judged by a photobleaching phenotype is shown above the bars. **e, f** Representative GC-MS chromatograms showing accumulation of intermediates in silenced plants (target gene plus *PDS*) in comparison to *PDS-GFP*-silenced plants. Chromatograms were normalised by leaf sample dry weight. Comparable data was obtained in two independent experiments. Compounds: 24-methylenecholesterol (**1**), 24-methyldesmosterol (**2**), ergosta-5,25-diene-3β,24ξ-diol (physalindicanol A) (**7**), ergosta-5,24(28)-diene-3β,25-diol (physalindicanol B) (**8**), (22*R*)-ergosta-5,24-diene-3β,22-diol (**10**), (22*R*)-ergosta-5,24(28)-diene-3β,22,25-triol (phyministerol A) (**12**), (22*R*)-ergosta-5,24-diene-1α,3β,22-triol (**14**), (22*R*)-ergosta-5,24(28)-diene-1α,3β,22,25-tetrol (**15**). Source data are provided as a Source Data file.

lack of activity. Strains were therefore cultured under modified conditions with larger culture vessels that offered better aeration. Nonetheless, we could still not detect the target lactone **16** in yeast. However, upon careful analysis of product profiles, we observed new peaks that were only present in yeast strains including all three CYP genes. One of them could be identified as (22*R*)-ergosta-5,24-diene-1α,3β,22,26-tetrol (**17**), the aglycone of cilistol v[77], by comparison to an authentic standard obtained by chemical reduction of withanoside V aglycone (**16**) (Fig. 7a, c). These results suggest that *Pp*CYP749B2 is indeed active in yeast and can oxidise C-26 of intermediate **14** but cannot complete the full oxidation to the lactone ring of **16** without support by other enzymes in yeast.

To find other enzymes that might complete the generation of the lactone moiety, we revisited the enzyme classes encoded by our withanolide gene clusters. As short-chain dehydrogenases/reductases are known to be capable of oxidising lactols to lactones, for example in the biosynthesis of nepetalactone[78], their involvement seemed highly plausible. Gratifyingly, yeast strains that contained one of the *SDR* genes from *W. somnifera* or *P. pruinosa* in addition to the three *CYP* genes successfully produced withanoside V aglycone (**16**) (Supplementary Fig. 24). In *N. benthamiana*, co-expression of *SDR* together with the *CYP* genes led to a 3-7-fold increase of the peak area of withanoside V aglycone (**16**) (Fig. 7d).

To get further insights into the biochemical interplay of CYP749B2 and SDR, we performed in vitro assays with both enzymes; microsomes enriched with *Pp*CYP749B2 were obtained from yeast, while two *Pp*SDR homologues a/b containing His-tags were produced in *E. coli* and purified by affinity chromatography. In vitro assays were carried out with the tetrahydroxylated intermediate **17** obtained by semisynthesis as a substrate (Fig. 7e). Positive controls containing both *Pp*CYP749B2 and *Pp*SDR-a/b showed formation of withanoside V aglycone (**16**) as expected, confirming that all enzymes were active under assay conditions. Reactions only containing *Pp*CYP749B2 did not result in formation of lactone **16**, matching to our results for yeast strains lacking SDR. Surprisingly, assays containing only *Pp*SDR-a without *Pp*CYP749B2 reproducibly showed small amounts of the lactone product **16**, but much less than when *Pp*CYP749B2 was also included. This suggests that SDR can not only carry out the last oxidation from a putative aldehyde **18a** or lactol **18b** intermediate to lactone **16** but also – much less efficiently than CYP749B2 – the prior oxidation from alcohol **17** to aldehyde **18a**.

Lastly, we performed virus-induced gene silencing in *W. somnifera* to support the role of *CYP87G1, CYP88C7, CYP749B2,* and *SDR in planta*. Fragments for silencing were designed to target all gene copies from the two sub gene clusters (Fig. 8a). Silencing constructs also contained a fragment for the phytoene desaturase (*PDS*) visual marker gene[79] to

serve as a guide for sample harvesting (Fig. 8a, b). Silencing of *PDS* alone caused a decrease of withaferin A levels compared to a non-specific control targeting the gene of green fluorescent protein (which is absent in *W. somnifera*) (Supplementary Fig. 25); hence, all *PDS* co-silencing experiments were compared to a *PDS-GFP* co-silencing negative control. Silencing of *24ISO* (together with *PDS*) was used as a positive control[14]. Successful silencing of target genes was also supported by qPCR (Fig. 8c). Silencing of all genes with the exception of *SDR* resulted in a drastic decrease of withaferin A levels to 4–22% mean values compared to the *PDS-GFP* silencing control; only for *SDR* no such decrease was observed (Fig. 8d). A similar metabolic effect was observed for two additional major compounds that were also classified as withanolides based on their high-resolution masses and retention times (Supplementary Fig. 26). This provided strong support that the withanolide gene cluster described here is indeed crucial for withanolide biosynthesis *in planta*.

To further corroborate the biosynthetic roles of the pathway genes, we searched for accumulation of pathway intermediates in GC-MS samples from silenced plants. *24ISO* silencing led to accumulation of 24-methylenecholesterol (**1**); *CYP87G1*-silenced plants accumulated 24-methyldesmosterol (**2**); plants with silenced *CYP88C7* contained (22*R*)-ergosta-5,24-diene-3β,22-diol (**10**); and *CYP749B2*-silenced plants were enriched in intermediate **14** (Fig. 8e, f). Notably, the same shunt products that we observed in *N. benthamiana* also accumulated during gene silencing; shunt products **7** and **8** were major products when any of the CYP genes was silenced; shunt products **12** or **15**, respectively, were detected when *CYP88C7* or *CYP749B2* were targeted. Accordingly, these metabolite profiles were in excellent agreement with our biosynthetic model derived from heterologous pathway reconstitution.

Taken together, our data provide support for a multistage oxidative assembly of the characteristic lactone ring of withanolides (Figs. 6a, 7a): First, C-22 is hydroxylated by CYP87G1; second, hydroxylation of C-1 is carried out by CYP88C7; lastly, the lactone ring is generated by step-wise oxidation of C-26 and lactonisation by CYP749B2 and SDR. CYP749B2 is crucial for the initial C-26 hydroxylation of intermediate **14** to intermediate **17**, while SDR is solely responsible for the final conversion of aldehyde **18a** or lactol **18b** to lactone **16**. In contrast, both CYP749B2 and SDR can carry out the oxidation of alcohol **17** to aldehyde **18a** or lactol **18b**. In *N. benthamiana* and *W. somnifera*, background enzymes can partially replace the dedicated withanolide gene cluster SDR, whereas in yeast the SDR is a crucial factor for successful lactone formation. In addition, shunt reactions by $\Delta^{24(25)}$ epoxidation can occur in the heterologous host *N. benthamiana* and the native producer *W. somnifera* alike. Based on these results, we propose the trivial names "withanolide biosynthesis 22-hydroxylase" (W22H) for CYP87G1, "withanolide biosynthesis 1-hydroxylase" (W1H) for CYP88C7, "withanolide biosynthesis 26-oxidase" (W26O) for CYP749B2, and "withanolide biosynthesis 26-dehydrogenase" (W26DH) for SDR. Together, these three CYPs and the SDR are therefore responsible for the biosynthesis of the key δ-lactone ring and for preparing the enone formation in the A ring of withanolides.

## Discussion

In this work, we identified a gene cluster for withanolide biosynthesis in Solanaceae plants. The gene cluster reported here adds to several previously reported cases of gene clustering in specialised metabolism in Solanaceae plants, for example for biosynthesis of glycoalkaloids, modified fatty acids, acyl sugars, or terpenoids[23,47,80,81]. As such, the Solanaceae family represents an increasingly intriguing model system to study gene clustering in plants. The function of the gene cluster described here for withanolide biosynthesis is supported by our phylogenomics analysis, the biochemical activity of CYP87G1, CYP88C7, CYP749B2 and SDR deduced by metabolic engineering in yeast and *N. benthamiana*, and virus-induced gene silencing in *W. somnifera*. Previously, several genes were already connected to withanolide biosynthesis. A CYP749 from *W. somnifera* identified by a transcriptome-based

approach was associated with withanolide biosynthesis based on virus-induced gene silencing, which resulted in a strong change in the withanolide profile[20]. Employing a similar approach, a recent study in *P. angulata* implied a role of a CYP749 and a CYP88 in the biosynthesis of physalin-type withanolides[82]. The biochemical functions of these CYPs as well as the genomic locations of the underlying genes have been unknown so far. We located these previously suggested candidates in our assembly of *W. somnifera*. Indeed, the closest homologues of these CYPs are encoded in the gene clusters described here (Supplementary Table 4). However, other genes identified in previous studies to be involved in withanolide production in *Withania somnifera* like WsCYP93Id[21], WsCYP71B10[20], glycosyltransferases[83] and others[84] were not part of or located close to the gene cluster described in this study (Supplementary Table 5). Some of these genes were moderately co-expressed with *24ISO* (Supplementary Table 6). Further work will be required to study the functional role of these genes in withanolide biosynthesis.

Similar to examples from other plants[85], our synteny analyses highlight the plasticity of plant biosynthetic gene clusters. Even in closely related species of the same genus, changes in gene order and orientation as well as gene duplications are not unusual. How these differences are relevant for increasing the structural diversity of withanolides will need to be addressed in the future. A unique feature of the withanolide gene cluster is the occurrence of sub gene clusters, as supported by synteny analysis, gene expression analysis (Supplementary Figs. 9–11), and biochemical demonstration of functional conservation (Supplementary Fig. 19). Gene duplications followed by neofunctionalization are common drivers of evolution[30]. In the case of withanolide biosynthesis, it appears that such an evolutionary event has occurred not only on the single gene level, but at a whole gene cluster level. Future studies will need to carefully address functional differences between the sub gene clusters to gain further insights into the evolution and biological role of withanolides in nightshade plants.

Elucidating and engineering the biosynthetic pathways of specialised steroids from plants still represents a major challenge. While many other compound classes such as alkaloids or other terpenoids are nowadays readily accessible in yeast or *N. benthamiana*, progress for steroids has been lagging behind. Only recently, the first successful metabolic engineering examples of steroidal natural products derived from the more common precursor cholesterol were published[86–89]. These have been facilitated by the fact that cholesterol natively accumulates in *N. benthamiana*, albeit at low levels. In contrast, our data shows that the production of relevant levels of 24-methylenecholesterol (**1**) or 24-methyldesmosterol (**2**) in yeast or *N. benthamiana* requires substantial additional engineering of sterol metabolism. In yeast, lowering the formation of shunt products arising from the misbalance of 24ISO and 7RED activity will be critical. The production of withanolide pathway intermediates in *N. benthamiana* appeared more efficient, even though an extensive set of genes had to be co-expressed. For maximum production of 24-methyldesmosterol (**2**) in *N. benthamiana*, we had to transiently overexpress 15 genes at the same time. Maximum production of withanoside V aglycone (**16**) was achieved by co-expression of 19 genes. To our knowledge, this setup is amongst the most complex examples reported to date[90–92]. Once again, this highlights the tremendous power of efficient and facile co-expression in *N. benthamiana* for gene discovery and pathway engineering in plants[66]. Our metabolic engineering efforts now enable further elucidation of withanolide biosynthesis, in order to close the gap from traditional medicine to modern drug development and production.

## Methods
### Plant materials

*W. somnifera* (XX-0-STGAL−22/1985) plants were grown in the Botanical Garden of TU Braunschweig in a greenhouse with natural light at approximately 20 °C. Genes from *P. pruinosa* were obtained as

synthetic genes from GeneWiz (Azenta Life Sciences) based on the published genome sequence[37] and our re-annotation without handling of the plant.

## Genome sequencing, assembly, and annotation

DNA extraction, quality control, short fragment depletion, and DNA repair for the Oxford Nanopore Technologies sequencing approach was performed with a CTAB-based protocol, Short Read Eliminator kit, and SQK-LSK109[93]. Sequencing was performed via MinION Mk1B on R9.4.1 flow cells. Basecalling was performed with Guppy v 6.4.6+ae70e8f in high accuracy mode. A genome sequence was constructed with NextDenovo2 v2.5.2 (read_cutoff = 1k, genome_size = 3 g, and seed depth = 30)[94]. RagTag v2.1.0[95] was used to reorder the contigs in the assembly based on homology with the close relative *P. pruinosa* genome assembly for direct comparison. Given the high N50 (71 Mb) of the assembly, nine contigs less than 100 kb in size were removed based on no BUSCO genes identified in those contigs. The reads were then mapped to the genome assembly using minimap2[96] to infer the ploidy and to estimate the genome size. Purge_haplotigs[97] was used to generate a coverage histogram to assess the ploidy of the assembly. Mapping-based Genome Size Estimation (MGSE)[98] was used to predict the genome size.

RNA was extracted and subjected to cDNA synthesis and sequencing (Supplementary Method 1). RNA-seq reads (Supplementary Table 7) were aligned with HISAT 2.2.1[99] using default parameters to generate hints for the gene prediction. A homology-based gene prediction of the *W. somnifera* genome was performed using GeMoMa v1.9[100,101], based on annotations of six Solanoideae sub-family species (*Physalis floridana*[36], *Datura stramonium*[34], *Iochroma cyaneum*[102], *Atropa belladonna*[34], *Lycium barbarum*[103], and *Solanum lycopersicum*[42]). For each of these six species, extracted coding sequences were aligned to the *W. somnifera* genome sequence with MMseqs2[104] (version 15-6f452) using default parameters suggested by GeMoMa. For predicting gene models, GeMoMa utilised protein-coding gene alignments, intron position conservation, and RNA-seq data for refining intron boundaries. The resulting six gene annotation sets were filtered and merged using GeMoMa Annotation Filter (GAF) with parameters f = "start = ='M' and stop = ='*' and (isNaN(score) or score/aa > =3.50)" and atf = "tie = =1 or sumWeight > 1". GeMoMa was also used to annotate the genome sequences of *P. pruinosa* and *P. grisea* using identical parameters, with a modified filtering step, f = "start = ='M' and stop = ='*' and (isNaN(score) or score/aa > =4.00)" and atf = "tie = =1 and sumWeight > 1" for both species. BUSCO v5.8.2[105] with the solanales_odb12 dataset was used to assess the completeness of the genome assemblies and annotations. The sources of genomic data sets are listed in Supplementary Table 8.

## Identification of transposable elements

For repeat sequence comparison, transposable elements (TEs) and tandem repeats were annotated in the genome sequences of *Withania somnifera* and *Physalis pruinosa* using Extensive *de*-novo TE Annotator[106] (EDTA) v2.2.1 and Tandem Repeats Finder (TRF)[107] v4.10.0, respectively, as described in Supplementary Method 2.

## Synteny analysis

JCVI/MCscan[108] v1.4.14 was employed to compare the local synteny among the genome sequences of known withanolide-producing species, *P. floridana*, *P. pruinosa*, *P. grisea*, *W. somnifera*, *D. stramonium* and *D. wrightii*. Non-withanolide producing species, *S. lycopersicum*, *S. tuberosum* and *N. tabacum* were also included. The sources of genomic data sets are listed in Supplementary Table 8. The region surrounding the *P. pruinosa 24ISO* gene was selected to identify microsynteny among these species. Connection of genes between species was validated and manually revised based on phylogenetic trees (Supplementary Fig. 3-8).

## Phylogenetic analyses

Phylogenetic trees were constructed using a total of 32 Solanaceae species (Supplementary Table 8). For each gene family, *Physalis pruinosa* gene sequences from the biosynthetic gene cluster were used as queries in BLASTp[109] searches against the polypeptide sequences of these 32 plant species, and the top BLAST hits were selected. This was performed using the Python script collect_best_blast_hits.py (https://github.com/bpucker/ApiaceaeFNS1)[110]. To differentiate between related gene families, additional outgroup sequences were included: *SSR1* and *SSR2* for *24ISO*; flavonoid biosynthesis genes *F3'H* and *F3'5'H* for *CYPs*; *DFR* for *SDRs*; and *FLS*, *F3H*, and *AOP* for *ODDs*. Additional plant sequences for *STs* and *ATs* were also incorporated into their respective trees. Sequences were aligned using MAFFT[111] v7.520 with default settings and the L-INS-i accuracy-oriented method. The amino acid alignments were translated back to codon alignments using pxaa2cdn from PHYX[112]. Alignments with occupancy below 10% were removed using pxclsq from PHYX[112]. The final phylogenetic trees were generated using IQ-TREE[113] v2.3.6 with --alrt 1000 -B 1000 --seqtype CODON. The best-fit evolutionary models were inferred using ModelFinder[114]: MG + F3X4 + I + R5 for 24ISO, MG + F1X4 + R8 for CYPs, MGK + F3X4 + R7 for SDRs, MG + F3X4 + R7 for STs and ATs, and MG + F3X4 + R8 for ODDs. Phylogenetic trees were visualised and annotated using iTOL[115] v6.8.

## Expression analysis

To evaluate the expression of genes from identified gene clusters in withanolide-producing species analysed in this study, publicly available paired-end RNA-seq datasets were collected. However, sufficient datasets with diverse tissue samples were only available for *D. stramonium* and *W. somnifera*. The FASTQ files for both species were retrieved from the Sequence Read Archive (SRA) using prefetch and fasterq-dump from NCBI SRA toolkit v3.1.0. Transcript abundance was quantified using Kallisto v0.50.1[116] based on the coding sequences. The individual count tables were then merged and filtered with customised Python scripts[110,117]. Expression heatmaps were generated in R to visualise the expression patterns of the identified gene clusters (https://github.com/NancyChoudhary28/Withanolide_biosynthesis/).

## Metabolic engineering of *S. cerevisiae*

For metabolic engineering of yeast (*S. cerevisiae*), the EasyClone-MarkerFree system was used[54]. All yeast strains generated in this study were derived from *S. cerevisiae* ST7574 (CEN.PK background) expressing *cas9* available on Euroscarf. Biobricks were assembled via USER cloning[118]. Amplification of all genes and promoters was performed using Phusion U polymerase (Thermo Fisher Scientific) with primers containing uracil overhangs. Resulting biobricks were assembled into AsiSI/Nb.BsmI (NEB) digested integrative vectors by USER cloning. Assembled synthetic constructs were transformed into NEB 5-alpha competent *E. coli* cells (NEB). Positive clones as judged by colony PCR were verified through Sanger sequencing (Microsynth Seqlab). Linear integration fragments containing the target genes flanked by homology arms for genomic integration were generated by NotI-HF (NEB) digestion. Linearised synthetic constructs were introduced into *S. cerevisiae* using the standard lithium acetate method[119]. Site-specific integrations were confirmed using standardised primers included in the EasyClone-MarkerFree Vector Set. Yeast strains generated in this study are listed in Supplementary Table 9, primers in Supplementary Data 4, plasmids in Supplementary Table 10, and coding sequences of inserted genes in Supplementary Data 5.

For gene deletions, homology-directed repair templates were prepared with High-Fidelity Q5 polymerase (NEB) using a PCR with 10 cycles and primers containing overlapping sequences. Guide RNAs (gRNAs) for deleting *ERG4*, *ERG5* and *24ISO* were designed using CHOPCHOP[120]. The required 20 nt gRNA sequences (Supplementary Table 11) were introduced into pCfB8792 (Addgene Plasmid # 126910)

by inverse PCR using primers containing gRNA sequences as overhang. The resulting amplicons containing modified gRNA sequences were gel purified (Macherey-Nagel) and treated with DpnI (NEB) for 30 min at 37 °C. DpnI-digested product was ligated using T4 DNA ligase (NEB) for 2 h at room temperature and transformed into NEB 5-alpha competent *E. coli* cells. Sequencing was done to confirm the gRNA sequence from isolated plasmids.

Cultures for yeast strains were set in triplicates for extraction of metabolites. Primary cultures were cultivated at 30 °C under continuous shaking at 210 rpm in YPD broth (Carl Roth) containing peptone (20 g/L), yeast extract (10 g/L) and glucose (20 g/L) supplemented with 200 mg/L G418. Overnight primary cultures were used to set secondary cultures with a starting $OD_{600}$ ~ 0.2 in YPD medium without antibiotics. Secondary cultures of synthetic constructs harbouring galactose-inducible promoters were induced with YP medium containing peptone (20 g/L), yeast extract (10 g/L) supplemented with filter-sterilised galactose (20 g/L). Secondary cultures were grown for 4 days and $OD_{600}$ was measured just before harvesting; then, metabolite profiles were analysed by GC-MS and LC-MS as described below.

### Transient expression in *Nicotiana benthamiana*

Transient expression in *Nicotiana benthamiana* was performed with *N. benthamiana* LAB strain[121], which was grown from seeds either in a greenhouse with 11 to 16 h of illumination per day at 21–23 °C or in a phytochamber with 16.5/7.5 h photoperiod at 100 µmol m$^{-2}$ s$^{-1}$ light intensity and a temperature of 22 °C during the day and 20 °C during the night. When possible, genes were amplified from cDNA (*W. somnifera* and *A. thaliana*) or alternatively obtained as synthetic genes (*W. somnifera* and *P. pruinosa)* and cloned into the plasmid pHREAC[122] by Golden Gate cloning or In-Fusion cloning as reported[121]. If necessary, Golden Mutagenesis[123] was used to remove undesired BsaI sites with silent mutations. All primers for metabolic engineering and transient expression in *N. benthamiana* are listed in Supplementary Data 4. The resulting plasmids were transformed into *Agrobacterium tumefaciens* GV3101 by electroporation. The *A. tumefaciens* strains were cultured in LB medium with antibiotics (25 µg/mL gentamicin, 50 µg/mL rifampicin, 50 µg/mL kanamycin) for 2 days at 28 °C with shaking. Cells were harvested, resuspended in MMA infiltration buffer (10 mM MgCl$_2$, 10 mM MES, 100 µM acetosyringone), and adjusted to $OD_{600}$ 0.1. For co-expression of multiple genes, the corresponding strains were mixed so that each strain had a final $OD_{600}$ 0.1. The mixtures were syringe-infiltrated into the abaxial side of 4-week-old *N. benthamiana* leaves. For compound purification, vacuum infiltration was used instead as described below. After infiltration, plants were maintained in a greenhouse or phytochamber until further analysis. Leaf samples were harvested 7 days post-infiltration for GC-MS and LC-MS analysis as described below.

### GC-MS sample preparation

GC-MS analysis of yeast strains was performed with cell pellets. Cells were harvested at 5000 × *g* for 5 min and washed twice with sterile water. Harvested cells (1.5 mL) were saponified at 90 °C for 10 min with 500 µL of 20% aqueous KOH in 50% ethanol containing 5 µg/mL of 5α-cholestane as internal standard. To the saponified extracts, 0.5 mm glass beads (Carl Roth, Germany) and either 500 µL hexane or 500 µL ethyl acetate were added depending on the expected polarity window of metabolites. Samples were then lysed in a FastPrep-24TM 5 G homogeniser (MP Biomedicals, CA, USA) with predefined settings for *S. cerevisiae*. The organic layer was transferred to a glass vial, and the extraction process repeated a second time. Both organic layers were pooled in a glass vial and concentrated *in vacuo* using an RVC 2-25 CDplus (Christ, Osterode am Harz, Germany) rotary vacuum concentrator with a cooling trap. Dried extracts were derivatised using 1:1

pyridine: (BSTFA + 1% TMCS) at 70 °C for 1 h. GC-MS analysis was carried out as described below.

For GC-MS analysis of *N. benthamiana* leaf samples after transient expression, 10 leaf disks were excised using a cork borer no. 5 (ø = 10 mm) and lyophilised until their weight remained constant (≈ 20 mg). Then, a 5 mm steel bead was added, and the disks were ground in a MM 400 ball mill (Retsch, Haan, Germany) at 30 Hz for 20 s. Ground leaves were saponified at 70 °C for 1 h with 500 µL of 10% aqueous KOH in 90% ethanol, containing 10 µg/mL of 5α-cholestane as internal standard. Afterwards, 300 µL deionised water was added. The saponified mixture was extracted twice with either 500 µL hexane or 500 µL ethyl acetate depending on the expected polarity window of metabolites. Before extraction with ethyl acetate, ethanol from the saponification solution was evaporated in a rotary vacuum concentrator with a cooling trap. After addition of solvent, samples were vortexed well and centrifuged for 1 min at 10,000 × *g*. Both organic layers were pooled in a glass vial and concentrated *in vacuo* as for yeast samples. Derivatisation was performed for 1 h at 70 °C using 75 µL of 1:1 pyridine: (BSTFA + 1% TMCS). Afterwards, the samples were diluted with 100 µL ethyl acetate and centrifuged for 10 min at 10,000 × *g*. The supernatant was subjected to GC-MS analysis as described below.

Semi-quantification of compounds from GC-MS data (shown in Fig. 6b) was performed by integration of peak areas from compound-specific extracted ion chromatograms (EIC). For each compound, the EIC of the most abundant ion was chosen. Compounds **2, 6, 7, 8, 10**, and **11** were extracted more efficiently with hexane and therefore quantified from hexane extracts. Ethyl acetate extracts were used for integration of compounds **12, 14**, and **15**. The following EICs were chosen for integration of each compound: 5α-cholestane (internal standard), *m/z* 217; **2**, *m/z* 129; **6**, *m/z* 363; **7**, *m/z* 157; **8**, *m/z* 157; **10**, *m/z* 295; **11**, *m/z* 293; **12**, *m/z* 183; **14**, *m/z* 383; **15**, *m/z* 183. Peak areas were normalised to internal standard and sample dry weight and converted into relative amounts with the highest mean value set to 100%.

### LC-MS sample preparation

LC-MS was used for the detection of oxidised withanolide pathway intermediates. *N. benthamiana* leaf samples and yeast cells were harvested in the same way as described for GC-MS analysis. 600 µL of ethyl acetate were added to the ground leaf powder or yeast cell pellets and vortexed vigorously. For quantification of withanoside V aglycone (**16**), 0.1 mg/mL emodin was added to ethyl acetate as an internal standard. The mixture was subsequently centrifuged at 14,000 × *g* for 15 min. The supernatant was transferred to a glass vial and concentrated *in vacuo*. Afterwards, the dried extracts of *N. benthamiana* leaf samples were redissolved in 600 µL ethanol, the dried extracts of yeast cell pellets were redissolved in 200 µL acetonitrile and analysed by LC-MS as described below.

### General chemical methods and chemicals

The controls and numbers of biological replicates used for GC-MS and LC-MS analyses are described in the figures and figure legends; no technical replicates were analysed. GC-MS analyses were carried out on a Hewlett Packard HP6890N GC system connected to a mass selective detector 5973 N with an OPTIMA 5 MS column (30 m × 0.25 mm i.d., 0.25 µm film, Macherey-Nagel, Düren, Germany). Helium was used as carrier gas at constant flow rate of 1.5 mL/min. Injection volume was 1 µL with a split of 1:5. For the quantification of 24-methyldesmosterol (**2**), the initial oven temperature was set to 100 °C followed by a ramp of 30 °C/min until 275 °C. Thereafter, the temperature was further raised to 290 °C with a ramp of 3 °C/min and held for 4 min, then raised to 300 °C with a ramp of 3 °C/min and held for 3.83 min. The total run time was 22 min. For all other measurements, the initial oven

temperature was set to 100 °C followed by a ramp of 30 °C/min until 275 °C. The temperature was then further raised to 300 °C with a ramp of 3 °C/min and held for 15.83 min. The total run time was 30 min. Mass spectra were obtained with a scan range of $m/z$ 43 to 800, with a solvent delay of 8 min after injection. GC-MS data was analysed using Agilent MSD ChemStation F.01.03.2357.

Analytical and semipreparative LC-MS measurements were performed on an Agilent Infinity II 1260 system consisting of a G7167A autosampler, G7116A column thermostat, G7111B quaternary pump, G7110B make-up pump, G7115A diode array detector, G1364F fraction collector, and G6125B single quadrupole mass spectrometer equipped with an ESI source (positive mode, 4000 V, 12 L min⁻¹ drying gas, 350 °C gas temperature). Analytical LC-MS measurements were carried out either using a C8 (data shown in Fig. 7b) or a C18 column (all other LC-MS measurements). C8 measurements were carried out using a Phenomenex Kinetex 2.6 μm C8 100 Å 4.6 × 150 mm column at 20 °C with 5 mM ammonium acetate in water (mobile phase A) and 5 mM ammonium acetate in MeOH (mobile phase B) and the following gradient: 0–5 min, 10–60% B; 5–12 min, 60–75% B; 12–15 min, 75–100% B; 15–21 min, 100% B; 21–21.1 min, 100–10% B; 21.10–24 min, 10% B. The flow rate was 0.8 mL/min. The injection volume was 5 μL and the total run was 24 min. C18 measurements were carried out using a Poroshell 120 EC-C18, 100 × 4.6 mm, 2.7 μm column at 40 °C with 0.1% (v/v) formic acid in water (mobile phase A) and acetonitrile (mobile phase B) and the following gradient: 0–8 min, 20–60% B; 8–10 min, 60–95% B; 10–12 min, 95% B; 12–12.10 min, 95–20% B; 12.10–14 min, 20% B. The flow rate was 1 mL/min. The total run was 14 min. LC-MS data was analysed using Agilent OpenLab CDS ChemStation C.01.10. For the quantification of withanoside V aglycone (**16**), peak areas were integrated using the following EICs: **16**, EIC 443; emodin (internal standard), EIC 271.

High resolution MS/MS measurements for identification of withanoside V aglycone (**16**) were carried out on a Vanquish LC (Thermo Fisher) using a Phenomenex Kinetex 2.6 μm C8 100 Å 4.6 × 150 mm column at 30 °C with a flow rate of 0.8 mL/min and 5 mM ammonium acetate in water (mobile phase A) and 5 mM ammonium acetate in MeOH (mobile phase B) as well as the following gradient: 0–5 min, 70–100% B; 5–5.1 min, 100–70% B; 5.1–7, 70% B. Mass spectra were obtained on an Orbitrap Q Exactive Plus mass spectrometer (Thermo Fisher) operated in PRM mode isolating the precursor with a $m/z$ of 443 with an isolation window of 1 $m/z$ followed by fragmentation at 10, 35 and 60 eV and detection of product ions at a resolution of 17,500. The AGC (automatic gain control) target and maximum injection time were set to 2e5 and 100 ms, respectively. The heated ESI (electrospray-ionisation) source was operated at 0 eV CID (collision-induced dissociation), sheath gas flow 60, auxiliary gas flow 17, sweep gas flow 4, spray voltage 3.5 kV, capillary temperature 288 °C, S-lens RF level 50.0 and aux gas heater 475 °C. High-resolution MS measurements for identification of withanolides in VIGS samples were performed using a Waters QTof Premier mass spectrometer connected to a Waters Acquity UPLC system.

Flash chromatography was performed on an Biotage Isolera One with columns and solvents as described in the Supplementary Information. All methods and gradients for compound purification are described in detail in Supplementary Table 12-17.

NMR spectra of isolated compounds were recorded using Bruker spectrometers operating at 400, 500, and 600 MHz for ¹H NMR, and at 100, 126, and 151 MHz for ¹³C NMR. Experiments were conducted at 298 K with the specified deuterated solvents. Chemical shifts (δ) were referenced relative to the residual solvent signal (CDCl₃: $\delta_H = 7.26$ ppm, $\delta_C = 77.16$ ppm; C₅D₅N: $\delta_H = 8.74$, 7.58, and 7.22 ppm, $\delta_C = 150.35$, 135.91, and 123.87 ppm) and expressed in ppm, with coupling constants reported in Hz. Analysis was conducted using TopSpin (version 4.1.3) or MestReNova (version 14.3.1).

An authentic sample of 24-methylenecholesterol (**1**) was provided as a kind gift by Prof. Dr. Hans-Joachim Knölker (TU Dresden, Germany). 24-Methyldesmosterol (**2**) was synthesised according to a procedure by Edwards et al.[124]. Withanoside V aglycone (**16**) was prepared by enzymatic hydrolysis of withanoside V (PhytoLab) as described below.

### Preparation of withanoside V aglycone (16)

Withanoside V aglycone (**16**) was prepared by enzymatic hydrolysis of withanoside V (PhytoLab) following a modified protocol by Matsuda et al.[76]. A solution of withanoside V (1.2 mg, 1.56 μmol) in sodium acetate buffer (0.4 mL, 0.2 M, pH 5) was treated with cellulase (from *T. reesei*, 4 μL, ≥ 700 U/g, ρ 1.1-1.3 g/mL). The mixture was stirred gently at 50 °C for 24 h and then subjected to an SPE column (Oasis PRiME HLB 1cc (30 mg)). After loading and washing with 5% MeOH (H₂O:MeOH 95:5), steroids were eluted with 4 mL of 75% acetonitrile (H₂O:acetonitrile 25:75). The eluate was concentrated under reduced pressure to yield a mixture of the desired aglycone of withanoside V (**16**) with remaining or partially deglycosylated starting material (1.0 mg). This mixture was then purified by semipreparative HPLC to yield 0.4 mg (0.90 μmol, 58%) of withanoside V aglycone (**16**). The identity of the isolated compound was confirmed by comparison of ¹H NMR spectral data with literature[125]. Semipreparative purification was achieved with a Phenomenex Luna 5 μm C8 100 Å 250 × 100 mm column and a gradient of 5 mM ammonium acetate in water (A) and methanol (B) at a flow rate of 4 mL/min: 0–4 min, 80–100% B; 4–8 min, 100% B; 8–8.1 min, 100–80% B, 8.1–10 min, 80% B.

### Purification of pathway intermediates heterologously produced in *N. benthamiana*

For compound purification, *N. benthamiana* plants were vacuum-infiltrated[126] in a 9.2 L ROTILABO desiccator (Carl Roth, Karlsruhe, Germany) connected to a MZ 2 NT membrane pump (Vacuubrand, Wertheim, Germany) at 30 mbar for 1 min. 120 and 90 plants were used for the isolation of *Pp*CYP87G1 and *Pp*CYP88C7 (shunt) products, respectively. Leaves were harvested 7 days post infiltration and lyophilised for 3 days until the dry weight remained constant. The dried leaves were ground at room temperature into powder with a blender and extracted with hexane for 24ISO and *Pp*CYP87G1 products and with ethyl acetate for *Pp*CYP87G1 + *Pp*CYP88C7 products. The crude extracts were concentrated *in vacuo* and purified by consecutive rounds of chromatography as described in Supplementary Tables 12–17. NMR spectra of isolated compounds are shown in Supplementary Figs. 27–66.

### Chemical synthesis

Synthetic procedures used in the context of determining the C-22 stereochemistry of (22*R*)-ergosta-5,24-diene-3β,22-diol (**10**) and for preparing (22*R*)-ergosta-5,24-diene-1α,3β,22,26-tetrol (**17**) are described in Supplementary Method 3. The full synthetic routes are shown in Supplementary Fig. 67. NMR spectra of synthetic compounds are shown in Supplementary Figs. 68–99. X-ray crystallographic data from this work is shown in Supplementary Table 18.

### Heterologous protein production

For in vitro assays, codon-optimised *Pp*CYP749B2 was produced in yeast (strain KMY95) together with a cytochrome P450 reductase (CPR) and obtained as a microsomal fraction. Microsomes were isolated from yeast strains BSY1[127] as a negative control and KMY95 (this work). Yeast strains were grown in YPD medium at 30 °C for 48 h; then, cells were harvested by centrifugation at 5000 × *g* at RT for 5 min, washed with 10 mL TEK buffer (50 mM Tris, pH 7.4, 1 mM EDTA, 0.1 M KCl), and centrifuged again with the same conditions. The supernatant was discarded and washed cells were then lysed with

glass beads ($\varnothing$ = 0.5 mm, 1/3 volume of lysate) with a bead homogeniser (FastPrep-24Tm 5 G) shaking at 6.0 m/s for 40 s with two cycles in 7 mL TEB buffer (50 mM Tris, pH 7.4, 1 mM EDTA, 0.6 M sorbitol). The volume was adjusted to 20 mL with TEB buffer and the cell debris was removed by centrifugation at 10,000 × $g$ at 4 °C for 10 min. Microsomes were isolated from the supernatant by ultra-centrifugation at 100,000 × $g$ at 4 °C for 1 h. Microsomes were finally resuspended in 800 μL of TEG buffer (50 mM Tris, pH 7.4, 1 mM EDTA, 20% glycerol), aliquoted to 200 μL per tube and stored at −80 °C until further use.

SDR proteins were produced in *E. coli* and purified by affinity chromatography based on a His tag. *PpSDR-a* and *PpSDR-b* sequences were cloned into the protein overexpression vector pET-28a(+) containing a C-terminal 6xHis tag (Supplementary Data 4). The vectors were transformed into *E. coli* BL21 (DE3) for heterologous expression. A single colony for each strain was picked from LB agar plate with 50 μg/mL kanamycin and cultured overnight at 37 °C with shaking at 250 rpm. Starter cultures were diluted 100-fold into 1 L LB cultures with 50 μg/mL kanamycin and cultured at 37 °C, with shaking at 250 rpm until $OD_{600}$ reached 0.6-0.8. Cultures were then induced with 0.1 mM IPTG and cultured at 18 °C with shaking at 250 rpm for 16 h. Cell pellets were harvested by centrifugation at 3200 × $g$ for 10 min and then resuspended in 10 mL ice-cold loading buffer [50 mM Tris-HCl, 500 mM NaCl, 5% glycerol, 50 mM glycine, 20 mM imidazole, cOmplete EDTA-free protease inhibitors (Roche), pH 8]. After sonication in an ice bath for 8 min (5 s on, 10 s off), cell debris was removed by centrifugation at 25,000 × $g$ for 20 min at 4 °C. The supernatant was transferred to a 15 mL centrifuge tube; then, 800 μL Ni-NTA agarose (Cube Biotech) were added and gently mixed on a rocking platform for 1 h at 4 °C. The mixture was centrifuged at 3200 × $g$ for 1 min at 4 °C, the supernatant was removed, and the Ni-NTA agarose was washed twice with 10 mL ice-cold loading buffer, then transferred to a 2 mL tube, washed three times with 800 μL elution buffer [50 mM Tris-HCl, 500 mM NaCl, 5% glycerol, 50 mM glycine, 500 mM imidazole, pH 8], and the supernatant was collected after centrifugation at 2400 × $g$ for 1 min at 4 °C. The eluates were combined and concentrated using an Amicon® Ultra centrifugal filter, 10 kDa MWCO. The final protein concentration was measured using NanoDrop.

### In vitro assays with *Pp*CYP749B2 and *Pp*SDR
Assays with yeast microsomes alone were performed in 250 μL total volume containing 25 μL yeast microsomes, 22 μM of the substrate (22*R*)-ergosta-5,24-diene-1α,3β,22,26-tetrol (**17**), and 2 mM NADPH in 50 mM Tris-HCl buffer, pH 7.5. In vitro assays with *Pp*SDR alone were performed in 250 μL total volume containing 76 μg/mL enzyme, 22 μM of **17**, and 2 mM NADP⁺ in 50 mM Tris-HCl buffer, pH 7.5. In vitro assays with yeast microsomes and *Pp*SDR together were performed in 250 μL total volume containing 25 μL yeast microsomes, 76 μg/mL *Pp*SDR-a or -b, 22 μM of the substrate **17**, 1 mM NADP⁺, and 2 mM NADPH in 50 mM Tris-HCl buffer, pH 7.5. All reactions were incubated at 30 °C with shaking at 400 rpm for 16 h.

All reactions were stopped by addition of 250 μL ethyl acetate. After vortexing and centrifugation, 200 μL of the ethyl acetate layer were collected. The extraction was repeated twice. Ethyl acetate layers were combined and dried *in vacuo*. The dried extracts were then redissolved in 200 μL of acetonitrile and analyzed by LC-MS on a C18 column as described above.

### Virus-induced gene silencing
Virus-induced gene silencing in *N. benthamiana* is described in Supplementary Method 4. To evoke virus-induced gene silencing in *W. somnifera*, a short gene fragment of 145-400 bp was selected for each target gene, based on the leaf-expressed homologues from *W. somnifera* gene cluster 1. To target all homologues of target genes in

the two gene clusters, VIGS fragments were chosen from conserved regions (Fig. 8a). Possible off-targets outside the two gene clusters and VIGS fragment efficiency were predicted using si-Fi[128]. Fragments were designed in this way for *CYP87G1*, *CYP88C7*, *CYP749B2* and *SDR*. A second fragment targeting *phytoene desaturase* (*PDS*) was chosen as a visual marker (Fig. 8a)[79]. As a positive control, the *24ISO* construct from Knoch et al.[14]. was adjusted to our *W. somnifera 24ISO* sequence. As a negative control, a *GFP* fragment was chosen with no potential targets in *W. somnifera*. All fragments were obtained as synthetic gene fragments including the required overhangs for cloning (Azenta Life Sciences or Twist Biosciences). VIGS fragments are listed in Supplementary Table 19.

Vectors used for VIGS were pTRV1 and pTRV2-MCS (pYL156[129]). Using In-Fusion cloning (Takara Biosciences), the *PDS* fragment was cloned into pTRV2-MCS linearised with EcoRI-HF and BamHI-HF to give pTRV2-*PDS*. A pTRV2-*GFP* vector was prepared in the same way as an extra negative control. The vector pTRV2-*PDS* was further linearised with KpnI-HF and XhoI and either the *GFP* or a target gene fragment inserted by In-Fusion cloning. After confirmation by Sanger sequencing, all resulting plasmids were transformed into *Agrobacterium tumefaciens* GV3101 by electroporation.

*W. somnifera* plants were grown in a phytochamber with 16.5/ 7.5 h photoperiod at 100 μmol m⁻² s⁻¹ light intensity and a temperature of 22 °C during the day and 20 °C during the night. *Agrobacterium* strains containing pTRV1 or a pTRV2-derivative were grown and harvested as described in section "Transient expression in *Nicotiana benthamiana*". Cells were resuspended in infiltration buffer and diluted to an $OD_{600}$ of 1.0. Cell suspensions containing pTRV1 and a pTRV2-derivative were mixed in a 1:1 ratio and syringe-infiltrated into the leaves of 3–4-week-old *W. somnifera* seedlings. For each vector combination, six replicates were infiltrated. First signs of photobleaching on *PDS*-silenced plants appeared after two to three weeks post infiltrations.

Leaf samples were harvested after 4-5 weeks post infiltration for expression and metabolite analysis. For plants infiltrated with a pTRV2-*PDS* derivative, white (photobleached) leaf areas were chosen if available. The number of biological replicates showing photobleaching as an indicator of successful silencing is provided in the respective figures. Six to ten leaf disks were excised with a cork borer no. 4 ($\varnothing$ = 8 mm), transferred to a 2 mL tube and frozen in liquid nitrogen. For metabolite analysis, samples from all six biological replicates were collected and lyophilised. For analysis of withaferin A levels, the samples were ground and extracted as described in section "LC-MS sample preparations". Ethyl acetate used during LC-MS sample preparation contained 0.1 mg/mL emodin as internal standard. LC-MS samples were analysed on a C18 column as described above. Withaferin A was identified based on an authentic standard. Peak integration was done using the following EICs: withaferin A (**A**), EIC 471; unidentified withanolide at 7.6 min (**B**), EIC 493; unidentified withanolide at 8.2 min (**C**), EIC 495; emodin (internal standard), EIC 271. Peak areas were normalised by internal standard and dry weight. To check for accumulation of biosynthetic intermediates, samples were processed as described in section "GC-MS sample preparations".

For expression analysis, samples from white leaves of three replicate plants were collected as described above for metabolite samples. Leaves were ground in a pre-cooled ball mill and RNA extracted using the GeneJET Plant RNA purification kit (Thermo Fisher Scientific). After gDNA digestion, 100 ng of RNA were used for cDNA synthesis with the RevertAid First Strand cDNA Synthesis Kit (Thermo Fisher Scientific). qPCR was performed using the 2× qPCRBIO SyGreen Mix Lo-ROX (Nippon Genetics) with a QuantStudio3 qPCR cycler (Thermo Fisher Scientific), according to the manufacturer's instructions. 1 μL of 1:10 diluted cDNA was used as template. Primers used during qPCR are listed in Supplementary Data 4. Data was analysed

using the $2^{(-\Delta\Delta Ct)}$ method[130] and expression levels calculated in relation to *EF-1a* and *PDS-GFP* control plants.

## Reporting summary

Further information on research design is available in the Nature Portfolio Reporting Summary linked to this article.

## Data availability

Sequences of genes from the withanolide gene clusters shown here are provided as a Supplementary Data 1. Raw sequencing reads generated in this study have been deposited in the European Nucleotide Archive (ENA) with accession ERP150021 under BioProject PRJEB64854. RNA-seq data has been deposited in the ENA with accession ERR13615536. The assembled genome sequence and the structural annotation of *W. somnifera* have been deposited in GenBank with the assembly accession code GCA_965601375 [http://www.ebi.ac.uk/ena/browser/view/GCA_965601375] and are also provided via LeoPARD [https://doi.org/10.24355/dbbs.084-202503200640-0]. Re-annotations of the *Physalis* genome sequences are provided via LeoPARD [https://doi.org/10.24355/dbbs.084-202409130931-0] and also via GitHub [https://github.com/NancyChoudhary28/Withanolide_biosynthesis/tree/main/data]. Source data are provided with this paper.

## Code availability

Scripts and commands of the Circos plot generation, heatmap construction, and input sequences used to construct phylogenetic trees are described and uploaded in Zenodo [https://doi.org/10.5281/zenodo.15632852] and GitHub [https://github.com/NancyChoudhary28/Withanolide_biosynthesis].

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

## Acknowledgements

We are thankful to Dr. Christiane Wittmann (Botanical Garden Duisburg-Essen) for providing information about XX-O-STGAL-22/1985. We gratefully acknowledge financial support by the Deutsche Forschungsgemeinschaft (FR 3720/7-1 to J.F.; PU 718/2-1 to B.P.; INST 187/741-1 FUGG to C.-P.W.). In addition, work in the group of J.F. was supported by the SMART BIOTECS alliance between the Technische Universität Braunschweig and the Leibniz Universität Hannover, supported by the Ministry of Science and Culture (MWK) of Lower Saxony. J.P is funded by China Scholarship Council (202208320109). A.A. is funded by a scholarship from the Egyptian Ministry of Higher Education (call 2019/2020). P.H. received financial support provided by the European Research Council (ERC Consolidator Grant "RadCrossSyn"), Deutsche Forschungsgemeinschaft (Heisenberg-Program HE7133/8-1), and Boehringer Ingelheim Stiftung (Plus 3 Perspectives Programme). This work was supported by the BMBF-funded de.NBI Cloud within the German Network for Bioinformatics Infrastructure (de.NBI) (031A532B, 031A533A, 031A533B, 031A534A, 031A535A, 031A537A, 031A537B, 031A537C, 031A537D, 031A538A). We thank Dr. Annika Stein and Stephan Rohrbach (Leibniz University Hannover, Germany) for preliminary work on this project and Dave Biedermann (Leibniz University Hannover, Germany) for synthesis of 24-methyldesmosterol. Prof. Dr. Hans-Joachim Knölker (TU Dresden, Germany) is gratefully acknowledged for providing authentic 24-methylenecholesterol as a kind gift. Umicore is acknowledged for a generous gift of metathesis catalysts. We thank Dr. David Nelson (University of Tennessee, USA) and the P450 nomenclature committee for naming of CYPs. We thank Thorsten Marschall (Botanical Garden of TU Braunschweig, Germany) and Annette Kaiser for excellent technical support in handling *Withania somnifera* propagation. We thank Dr. Gerald Dräger (Leibniz University Hannover, Germany) for excellent X-ray crystallography support. We thank Dr. Lorenzo Caputi and Dr. Klaus Gase (Max Planck Institute for Chemical Ecology, Jena, Germany) for helpful advice regarding silencing construct design. We also thank all members of the research group Plant Biotechnology and Bioinformatics (TU Braunschweig, Germany) for discussion and support.

## Author contributions

J.F. and B.P. conceived and supervised the project. S.E.H., N.C., K.M., J.P., A.B., A.A., B.P. and J.F. designed the experiments. N.C. and B.P. performed the assembly and annotation of the *W. somnifera* genome sequence. N.C. performed re-annotations of *Physalis* genome sequences, synteny analyses, phylogenetic analyses, and gene expression analyses. K.M. designed and conducted yeast metabolic engineering experiments. J.P. designed and conducted metabolic engineering in *N. benthamiana*. S.E.H, J.P., A.B. and J.E. performed transient expression in *N. benthamiana*. S.E.H. and J.P. purified compounds. A.A. performed NMR analysis. S.E.H., K.M., J.P. and A.B. generated and interpreted chromatographic and mass spectrometric data. S.E.H. optimised and performed deglycosylation of withanoside V. J.P. performed heterologous protein production and in vitro assays. A.B. performed gene silencing in *N. benthamiana* and *W. somnifera*. R.F. performed high molecular weight DNA extraction and nanopore sequencing. C.-P.W. and M.H. designed and performed MS/MS and high-resolution MS measurements. M.B. and P.H. designed synthetic routes. M.B. performed synthesis. S.E.H., N.C., K.M., J.P., A.B., A.A., B.P. and J.F. wrote the manuscript. All authors reviewed and approved the final manuscript.

## Funding

## Competing interests

The authors declare no competing interests.
