## [Peer Review file · Nature Communications]

Phylogenomics and metabolic engineering reveal a conserved gene cluster in Solanaceae plants for withanolide biosynthesis

Corresponding Author: Professor Jakob Franke

Version 0:

Reviewer comments:

Reviewer #1

(Remarks to the Author)

The manuscript from Hakim, Choudhary, Malhotra, Peng et al is an impressive piece of work, and a great example of genomics-based elucidation of specialized metabolism as well as the utility of *Nicotiana benthamiana* as a tool. To summarize – the researchers sequence a genome, identify a conserved gene cluster, engineer chassis for production of withanolides and then characterize three new P450 enzymes.

In particular, I would like to praise the chemical elucidation work, which is incredibly well reported and thorough. There is an attention to detail present here that is unmatched in this field – notably the bravery to interpret EI spectra, perform complex syntheses, the commitment to isolating products, and the careful consideration of stereochemistry. Furthermore, I think the effort and varying methods to boost precursor production in both yeast and benth was good. The reporting of engineering methods that did not work is commendable. The high-quality genome sequence and decision to reannotate other genomes is also a good aspect of this manuscript.

Unfortunately, due to its scope, multidisciplinary nature, inclusion of high levels of detail and possibly due to four first-authors, the ultimate result ends up patchy with an incoherent narrative and I recommend it being revised before it is ready for publication.

Major points

The phylogenomics is under-developed. A cluster has been identified across multiple genomes and subject to synteny analysis. The authors also note that “The sub-gene cluster separation was also examined by phylogenetic analyses”. Yet the only conclusion or interpretation of the data I can see here is that it “strongly suggest that withanolide biosynthesis involves a conserved gene cluster”. Conserved in what way? Across multiple genomes? So, you are stating that there is a common origin (i.e. shared in an ancestor)? Also – how do the subclusters relate to each other? Did a single cluster duplicate into the 2x subclusters? There is no interpretation of the phylogenies that I can see – how do the gene trees exactly support a “conserved” cluster that duplicated? Would you infer an origin in Solanoideae then loss in *Solanum*? Also, the *Datura* genes are curious as the gene trees indicate both a and b genes tend to sit separately from the *Withania* and *Physalis* a+b genes perhaps indicating an independent duplication in that lineage. I would also recommend nucleotide or codon-based trees here – when the genes are so closely related / recently diverged, then you need more information to infer the phylogenies – the synonymous substitutions add lots more information to help resolve phylogenies. Either add more analysis with improved trees or be a bit clearer about what the current phylogenomics data is really showing.

I do not think that CYP749B2 has convincingly been shown to be withanolide lactone synthase. I would expect a CYP to be able to make the lactol here (hydroxy then hydrate/aldehyde then hemiacetal ring closure) but the oxidation of this lactol is not typically a CYP reaction. I would expect that native SDRs in benth are oxidizing it to the lactone, and that one of the SDR candidates in the cluster is performing the oxidation in the native pathway. There are some options to resolve this: note that this is a possible situation due to background benth enzymes; test to see if the co-expressing SDR candidate boosts production; perform appropriate in vitro reactions; look for the lactol product. For all the intricate analysis and characterization of shunt products, it is a bit disappointing that this a major result is not convincingly demonstrated – we have

a single small peak on Fig 6C and a spectrum 6D. I do think the compound ID is correct but the enzyme activity needs a little more work.

The paper is slightly unbalanced with an unusual narrative. There are three papers trying to emerge – one on gene clusters /phylogenomics, one on engineering precursor production and one on gene discovery. They don't completely sit together. For example, the genome sequenced is *Withania somnifera* but the biochemistry is all *Physalis pruinosa*. I assume the use of the word “conserved” for the gene cluster is a justification that they are not testing the *Withania somnifera* genes. Yet the *Physalis pruinosa* genome was already published and the cluster would have been obvious without phylogenomic analysis (though I acknowledge it was reannotated here). So, the gene discovery didn't need the *Withania somnifera* genome. Also there are discrepancies between the tree and the gene discovery - CYPs have proper names in the Fig 6 (e.g. PpCYP87G1) but the tree (SF6) has different names (PpCYP87G-b and PpCYP87G-a) etc. Sequence identity data would actually be useful to decide whether *Physalis pruinosa* and *Withania somnifera* have clear orthologs such that they are essentially the same enzyme. The abstract (which doesn't mention *Physalis pruinosa*!) makes us think that CYP87G1, CYP88C7, and CYP749B2 are from *Withania somnifera* – but would the *Withania somnifera* genes have the same names? How similar are they? Based on the tree and the naming it is not even clear which are the orthologs of the ones being tested here.

There are other groups working on the same cluster (<https://www.biorxiv.org/content/10.1101/2024.10.02.615186v2>). They focus on the evolution and regulation of the cluster. To distinguish the papers, and to aid in writing a more focused manuscript, perhaps the work here could be refocused on *Physalis pruinosa* biochemistry plus the synthetic biology. The *Withania* genome and phylogenomics could be less of a focus. If the authors can convincingly show withanolide lactone synthase then this would elevate the paper further and make it suitable for publication here.

Minor points

General critique is a lack of information in figures and figure legends. The plots need to be interpretable by non-experts and provide sufficient information.

Fig 3 need contig/chrom numbers and locations of the genes/clusters. Perhaps a number showing start and end of syntenic region. Is it a species tree on the LHS?

Generally for LCMS plots – what is the y axis and for TIC/EIC how is it normalized? E.g. Fig4b, 5bc, 6c. The Relative levels of 6b are a little confusing too – yeast and benth are plotted together but they are normalized separately? This needs a little more clarification in the legend. Could we get some information about the peak sizes overall? Even without standards, knowing whether we are at 10^3 or 10^6 is quite informative.

line 152 – what does “phylogenetically related” mean? Closely related would be better or “within the same subfamily” etc.

Genomics – supplementary table S1 – can you show full BUSCO (complete, dup, frag, missing) data?

Clarify line 124 – did you also reannotate TEs/repeats?

Condense some text – e.g. paragraph at 178-197

(Remarks on code availability)

Reviewer #2

(Remarks to the Author)

The work by Hakim et al., titled “Phylogenomics and metabolic engineering reveal a conserved gene cluster in Solanaceae plants for withanolide biosynthesis” is an attempt to understand the genetic organisation of genes involved in withanolide biosynthesis in Solanaceae plants. In this effort, authors have performed genome sequencing and assembly of *Withania somnifera* and compared it with genomes of withanolide producing solanaceae plants. From this analysis, they found a gene cluster and sub-clusters associated with 24ISO, which has been previously shown to form 24-methylene desmosteraol, a possible precursor for withanolides biosynthesis. Further, authors have shortlisted 3 genes of *Physalis pruinosa* and functionally co-expressed them with 24ISO in yeast and tobacco along to study their role in withanolide formation. Through the analysis of the products formed by the engineered yeast and tobacco, they claim a pathway for withanoside V aglycone formation. The study is interesting and the efforts are worthwhile in the backdrop of the current lack of understanding of withanolide biosynthetic pathway. Nevertheless, this study lacks focus and has several shortfalls for publication as listed below;

1. The major concern of this study is lack of in planta characterization of genes within the host plant. The use of only heterologous system is not an authentic way of deciphering the pathway, it needs to be established in host plant also. Since virus-induced gene silencing and overexpression methods are well established in *W. somnifera*, authors should have adopted these approaches to prove their claims.
2. As per the abstract, it sounds like authors have characterized genes from *W. somnifera*. However, in the results gene sequences of *P. pruinosa* have been used for expression in yeast and tobacco model. It is not comprehensible given the fact

that efforts were focussed genome analysis of *W. somnifera*, the most well-known plant for withanolides accumulation.

3. The authors have used 24ISO as an anchor to identify gene clusters, however there are several genes that are known to affect withanolide production in *W. somnifera*. How do those genes relate to this gene cluster? Do they have clusters of their own? Are these clusters expressed upon treatment to elicitors like methyl jasmonate?.
4. Figure 3 can be replaced with supplementary Fig 1. Do authors know the size of these gene clusters?. If so, needs to be elaborated and a scale should be included in the corresponding figures. The appearance of two gene clusters in *W. somnifera* is quite intriguing and can be discussed more. Was a sub-gene cluster found in *D. stamonium*?
5. There has been abundant work done already to improve triterpene and sterol production in yeast. The authors should review the literature about engineering yeast. Yang et al., (<https://doi.org/10.3390/biom11111710>) for instance has developed a yeast strain capable of producing ~200 mg/L of 24-methylene cholesterol, for this very purpose. Building upon this work would have resulted in a more favourable yeast testing platform. The bottle-neck caused by 7RED could be overcome by screening additional DWF enzymes as done by Yang et al (2021). Interestingly the expression of 24ISO resulted in shunt products while these were not seen in the original paper (Knoch et al., 2018), any explanation why?.
6. What is most striking is why the authors did not overexpress HMG-CoA reductase, a known rate limiting enzyme of the sterol pathway in yeast. This was later done for *N. benthamiana* but not for yeast! It is surprising that the overexpression of UPC2-1 did not improve sterol production in yeast. Authors should confirm its overexpression using qPCR and western blotting.
7. As the authors themselves have stated, the transient overexpression of 15 gene using mixed cultures of *Agrobacterium* is questionable as the level of induction achieved is not always the same. Generation of transgenic tobacco (which has well established transformation protocol) with enhanced substrate production would have been a more robust approach. In summary, the authors have heavily relied on the findings of Knoch et al (2018) for this work. Though there is evidence to suggest other phytosterols as the precursor of withanolides, the authors have chosen 24-methylene desmosterol for their genes which have limited their findings. Apart from the genomic resources generated, the engineering of model platforms and characterization of putative genes of the withanolide biosynthetic pathway are still in its nascent stages.

Minor points:

The trivial names can be reworked. In their current form, they sound like amino acid substitutions (W22H) rather than catalytic activity. Withanolide 1-hydroxylase is not appropriate as the substrate is not a withanolide.

The supplementary material is too long and complex. The raw spectral data can be shifted to the bottom or to an additional file. Several figures can be clubbed into a single figure.

(Remarks on code availability)

Reviewer #3

(Remarks to the Author)

The manuscript by Samuel Edward Hakim et al. employs a phylogenomics approach to identify a conserved gene cluster involved in withanolide biosynthesis and validates the functions of these genes by establishing a genetic engineering system in yeast and tobacco. I read this manuscript with great interest and commend the authors for their extensive efforts on this project. As my expertise lies in genomics rather than biosynthesis, my comments will focus on the genome assembly and phylogenomics sections.

1. I noticed that a genome of *Withania somnifera* has already been released in NCBI (https://www.ncbi.nlm.nih.gov/datasets/genome/GCA_039654485.1/). A comparison with this NCBI-released genome would be necessary to demonstrate the quality of the genome assembled in this study.

2. There is limited information regarding this genome reported in the results. For example, what is the ploidy level, estimated genome size, and heterozygosity of this genome? Additionally, how many ONT reads were sequenced for de novo assembly, and what is the average read length?

3. Given that the genome was assembled using ONT reads, which typically contain a high level of sequencing errors, the resulting genome may have a significant proportion of errors. This could lead to incorrect gene annotations, including premature stop codons and frameshifts. I suggest that the authors de novo assemble the RNA-seq reads into unigenes using the Trinity de novo package and compare these transcripts with their gene annotations. This would provide a clearer overview of the quality of the gene annotation.

4. The synteny analysis among different species is quite interesting. The results indicate that no 24ISO orthologue was found in *S. lycopersicum*, *S. tuberosum*, and *N. tabacum*. I am curious about the origin of this gene (24ISO). Did it arise from horizontal gene transfer (HGT), de novo formation from the genome sequence, or was it simply lost in these three species?

5. In line 162, what do the authors mean by "all conserved genes"? Are these genes within the same syntenic block? How many of these genes are there? Providing this information would help readers better understand the context.

6. The sentence in lines 168-169, "A few withanolides bearing ..." seems unrelated to the preceding context. Please clarify or revise this section.

7. Lines 178-186 read more like a discussion. I recommend moving this portion to the discussion section.

8. In lines 199-204, the authors need to provide more details regarding gene expression. How many genes and what samples/tissues were examined? The authors mention observing two clearly separated groups of expression patterns but do not provide details about these groups. How many genes are in each group, and what are their characteristics? Are they highly expressed in some tissues while low in others? Additionally, I strongly suggest that the authors sequence some Hi-C reads, as this could assist in genome scaffolding and help investigate the 3D structure of the genome. This may provide insights into why some of these genes co-express, possibly indicating they are in the same chromatin loop.

9. In line 298, where it states “only 24ISO,” I recommend adding the Latin name for the gene for clarity.

(Remarks on code availability)

Version 1:

Reviewer comments:

Reviewer #1

(Remarks to the Author)

The revisions required for this manuscript were a lot including new gene discovery across multiple systems, silencing, new phylogenetics, and more. The authors conducted all of these in a timely manner with high-quality results. I am pleased they were able to identify an SDR as predicted. Overall the work is a very good contribution to the literature demonstrating high quality work, worthy of publication. ...And just in time as Jing-Ke Weng's excellent pre-print emerged during the review process. (Timing is important in this competitive area with respect to the SDR hypothesis- my review was submitted 21 Oct 2024 and the Jing-Ke Weng preprint emerged on Christmas Day 2024.)

(Remarks on code availability)

Had a quick look at the code especially Expression_Heatmap.R and it looks suitable. The code is available and annotated and nicely documented. I did not try to run it.

Reviewer #2

(Remarks to the Author)

The revised manuscript, titled “Phylogenomics and Metabolic Engineering Reveal a Conserved Gene Cluster in Solanaceae Plants for Withanolide Biosynthesis,” represents a significant advancement from its previous iteration. I commend the authors for thoughtfully incorporating the critical suggestions provided by me and the other reviewers, meticulously conducting the necessary experiments, and diligently revising their manuscript. A particularly noteworthy contribution is the discovery of the role of SDRs in the biosynthesis of withanoside V aglycone, which substantially enhances the manuscript's impact. However, the manuscript still appears disjointed, lacking a cohesive narrative that guides the reader seamlessly through the content. To enhance clarity, every mention of CYP should be accompanied by an abbreviation indicating the species referenced—either “Ws” or “Pp.”

The engineering of the yeast platform for 24-methylenedesmosterol production could have been approached with greater deliberation. The authors sidestep deeper analysis by stating that “sterol metabolism in yeast underlies further regulatory constraints.” However, these constraints are well documented in the literature and have been successfully addressed. For instance, while the authors attempted a simple overexpression of HMGR, which led to a substantial increase in squalene, it failed to enhance 24-methylene desmosterol production. It is well established that sterol accumulation represses downstream genes such as ERG1. Overexpressing these downstream genes could have facilitated higher 24-methylene desmosterol yields.

Additionally, UPC2.1 serves as a master regulator for multiple genes, including those involved in precursor utilization. The studies cited by the author in their response have employed UPC2.1 overexpression without simultaneously downregulating competing pathway genes. A notable quote from their cited paper states that “the combination of downregulating ERG9 and overexpressing upc2-1 increased amorhadiene production” (<https://doi.org/10.1038/nature04640>). In this study, the competing gene, ERG5 was knocked out, which raises an important question: Why did UPC2.1 overexpression not influence 24-methylene desmosterol production? The authors should provide a more convincing explanation for this observation.

I strongly encourage the authors to inform readers about the mevalonate pathway engineering in yeast that failed to increase 24-methylene desmosterol production, as this would help prevent unnecessary duplication in future research efforts.

Figures 6a and 7a could be combined into a separate figure at the end, illustrating a well-constructed (proposed) pathway from 24-methylene desmosterol to withanoside V aglycone, including its shunt products, in yeast and *N. benthamiana*. This would unify the study and effectively convey the role of each CYP and SDR in the pathway.

A key question raised in my previous review remains unresolved: Do the enzymes studied by the authors establish the pathway under laboratory conditions, or is this the naturally occurring pathway in plants? This was the primary reason for recommending in planta characterization studies via VIGS and overexpression. The authors have conducted VIGS on the equivalents of the three CYP and SDR genes in *W. somnifera*, but they have co-silenced PDS as a visual marker for gene silencing. Consequently, this only validates the role of the CYP/SDR genes within a PDS-silenced background. Given that PDS silencing is known to have collateral effects on the metabolome, the CYP and SDR genes should be silenced independently of PDS to avoid confounding results.

The authors have exclusively quantified withaferin A levels in the silenced plants, but this raises an important question—are they suggesting that withanoside V aglycone acts as a precursor to withaferin A? Clarification on this point is necessary.

A more comprehensive investigation into the fate of other withanolides and withanosides in the silenced plants would greatly enhance the study's depth. Given that many withanolides co-elute with withaferin A in the same LC chromatogram, their quantification should be straightforward. The authors must extend their analysis to include these additional withanolides and provide a thorough discussion of their findings to shed light on the underlying metabolic dynamics.

The legend in Figure 8 should explicitly mention the names of compounds 1–15 for clarity.
Line 517: Sufficient acknowledgement to reviewer 1 should be given here.

(Remarks on code availability)

Reviewer #3

(Remarks to the Author)

The authors have addressed all my concerns and I have no further comments.

(Remarks on code availability)

Version 2:

Reviewer comments:

Reviewer #2

(Remarks to the Author)

Authors have tried their best to respond to the concerns raised. However, I still have my reservations on the following;

1. Use of VIGS with PDS background is not acceptable. There are already several published papers reporting the use of VIGS without PDS marker, but with a visual curly or viral infection phenotype in leaves.
2. Analysis of only withaferin A and few other unidentified withanolides is still a concern. For this also, there are several papers that have reported the quantification of major withanolides if not all withanolides.

(Remarks on code availability)

Point-by-point response

Please find enclosed our revised manuscript NCOMMS-24-55032. We are pleased about the positive nature of the reviewers' comments. We now carefully addressed all points raised by the reviewers. All substantial changes in our manuscript and SI have been highlighted in yellow. Major changes are:

- 1) We **identified and characterised a further biosynthetic gene (SDR)** that is essential for lactone formation in yeast, but not in plants. The function of *SDR* was studied by heterologous expression in *N. benthamiana* and yeast as well as *in vitro*. This is now shown in a new **Fig. 7**.
- 2) We performed **virus-induced gene silencing in *W. somnifera*** which supports that our gene cluster is indeed pivotal for withanolide biosynthesis *in planta*. This is shown in a new **Fig. 8**.

Our specific responses to the reviewer requests are:

Reviewer 1:

The manuscript from Hakim, Choudhary, Malhotra, Peng et al is an impressive piece of work, and a great example of genomics-based elucidation of specialized metabolism as well as the utility of Nicotiana benthamiana as a tool. To summarize – the researchers sequence a genome, identify a conserved gene cluster, engineer chasses for production of withanolides and then characterize three new P450 enzymes.

In particular, I would like to praise the chemical elucidation work, which is incredibly well reported and thorough. There is an attention to detail present here that is unmatched in this field – notably the bravery to interpret EI spectra, perform complex syntheses, the commitment to isolating products, and the careful consideration of stereochemistry. Furthermore, I think the effort and varying methods to boost precursor production in both yeast and benth was good. The reporting of engineering methods that did not work is

commendable. The high-quality genome sequence and decision to reannotate other genomes is also a good aspect of this manuscript.

Unfortunately, due to its scope, multidisciplinary nature, inclusion of high levels of detail and possibly due to four first-authors, the ultimate result ends up patchy with an incoherent narrative and I recommend it being revised before it is ready for publication.

Response: We thank the reviewer very much for their positive and constructive comments. We improved the narrative as described below in more detail to make the manuscript more coherent.

The phylogenomics is under-developed. A cluster has been identified across multiple genomes and subject to synteny analysis. The authors also note that “The sub-gene cluster separation was also examined by phylogenetic analyses”. Yet the only conclusion or interpretation of the data I can see here is that it “strongly suggest that withanolide biosynthesis involves a conserved gene cluster”. Conserved in what way? Across multiple genomes? So, you are stating that there is a common origin (i.e. shared in an ancestor)? Also – how do the subclusters relate to each other? Did a single cluster duplicate into the 2x subclusters? There is no interpretation of the phylogenies that I can see – how do the gene trees exactly support a “conserved” cluster that duplicated? Would you infer an origin in Solanoideae then loss in Solanum? Also, the Datura genes are curious as the gene trees indicate both a and b genes tend to sit separately from the Withania and Physalis a+b genes perhaps indicating an independent duplication in that lineage. I would also recommend nucleotide or codon-based trees here – when the genes are so closely related / recently diverged, then you need more information to infer the phylogenies – the synonymous substitutions add lots more information to help resolve phylogenies. Either add more analysis with improved trees or be a bit clearer about what the current phylogenomics data is really showing.

Response: We thank the reviewer very much for their valuable suggestions. We agree with the reviewer that the evolutionary history of the withanolide gene cluster seems to be very complex and will require detailed further study. As suggested by the reviewer, we therefore toned down the phylogenomics part and simultaneously clarified what our current phylogenomics analysis shows:

- Most importantly, we now clarify that “conserved gene cluster” refers to functional conservation of biochemical activity. We provide evidence that enzymatic function is conserved within species, i.e., between the two sub gene clusters in *W. somnifera* (with the only exception of CYP749B2), and between species (i.e., *W. somnifera* and *P. pruinosa*). This is shown in **Supplementary Fig. 19** and a newly added **Fig. 7**.
- We also replaced our previous phylogenetic trees by codon-based trees as suggested by the reviewer (**Supplementary Fig. 3-8**). Phylogenetic analyses of the identified genes confirmed that these genes from withanolide-producing species clustered together forming a distinct clade and no orthologs are found in non-withanolide producing species like tomato, potato, and *Nicotiana* (**Supplementary Fig. 3-8**).

The results and discussion sections have been adjusted to reflect these changes. For example, we added this statement about our phylogenetic analysis:

New version: Phylogenetic analyses of the identified genes revealed that these genes from withanolide-producing species clustered together, forming a distinct clade; no orthologues are found in non-withanolide producing species like tomato, potato, and *Nicotiana* (Supplementary Fig. 3-8).

I do not think that CYP749B2 has convincingly been shown to be withanolide lactone synthase. I would expect a CYP to be able to make the lactol here (hydroxy then hydrate/aldehyde then hemiacetal ring closure) but the oxidation of this lactol is not typically a CYP reaction. I would expect that native SDRs in benth are oxidizing it to the lactone, and that one of the SDR candidates in the cluster is performing the oxidation in the native pathway. There are some options to resolve this: note that this is a possible situation due to background benth enzymes; test to see if the co-expressing SDR candidate boosts production; perform appropriate in vitro reactions; look for the lactol product. For all the intricate analysis and characterization of shunt products, it is a bit disappointing that this a major result is not convincingly demonstrated – we have a single small peak on Fig 6C and a spectrum 6D. I do think the compound ID is correct but the enzyme activity needs a little more work.

Response: We thank the reviewer very much for suggesting an involvement of the SDR. We carefully examined this possibility by multiple approaches and can now confirm that the SDR is indeed involved in lactone formation based on multiple lines of evidence in yeast, in *N. benthamiana*, and *in vitro*. These new data are shown in a new **Fig. 7** that describes the formation of the lactone ring in detail:

- 1) Inclusion of *SDR* for co-expression in *N. benthamiana* led to a 3-7-fold increase in the amount of withanoside V aglycone (**16**) formation. This data is now included in **Fig. 7d**.
- 2) In yeast, we could previously not detect withanoside V aglycone (**16**) when our three *CYP* genes were co-expressed. We now re-analysed the yeast strains in more detail and identified the partially oxidised C-26 alcohol intermediate **17** based on a semisynthetic standard. These data show that CYP749B2 can oxidise C-26 but alone cannot form the lactone ring in yeast. This is now shown in **Fig. 7c** and **Supplementary Fig. 24**.
- 3) Adding *SDR* genes to yeast led to the successful production of withanoside V aglycon (**16**), confirming that SDR is a critical factor for lactone formation in yeast. This is shown in **Supplementary Fig. 24**.
- 4) *In vitro* assays with CYP749B2 and SDR with semisynthetic alcohol **17** as a substrate indicated that *PpSDR-a* alone can form the lactone in trace amounts, but efficient lactone formation requires the combination of CYP749B2+SDR combined (**Fig. 7e**).

We also performed VIGS of *SDR* as requested by Reviewer 2. While silencing was successful as judged by qPCR (**Fig. 8c**), we did not observe a decrease

of withaferin A levels when silencing *SDR*, whereas silencing of the three *CYP* genes or *24ISO* led to a very strong decrease (**Fig. 8d**). This result is in line with our previous result in *N. benthamiana* that *SDR* is not essential for lactone formation *in planta*.

From our data, we now firmly conclude that lactone formation is dependent on *SDR* activity, as kindly suggested by the reviewer. In *N. benthamiana* (and in *W. somnifera*), background enzymes can substitute the dedicated *SDR* encoded by the gene cluster to a certain degree, which led to a misinterpretation of our data in the original manuscript. We have now updated the manuscript in several instances to include the function of the *SDR*, most importantly in **Fig. 7**. We also highlighted the valuable contribution of the reviewer regarding the role of the *SDR* in the acknowledgement section.

The paper is slightly unbalanced with an unusual narrative. There are three papers trying to emerge – one on gene clusters /phylogenomics, one on engineering precursor production and one on gene discovery. They don't completely sit together. For example, the genome sequenced is Withania somnifera but the biochemistry is all Physalis pruinosa. I assume the use of the word "conserved" for the gene cluster is a justification that they are not testing the Withania somnifera genes. Yet the Physalis pruinosa genome was already published and the cluster would have been obvious without phylogenomic analysis (though I acknowledge it was reannotated here). So, the gene discovery didn't need the Withania somnifera genome. Also there are discrepancies between the tree and the gene discovery - CYPs have proper names in the Fig 6 (e.g. PpCYP87G1) but the tree (SF6) has different names (PpCYP87G-b and PpCYP87G-a) etc. Sequence identity data s would actually be useful to decide whether Physalis pruinose and Withania somnifera have clear orthologs such that they are essentially the same enzyme. The abstract (which doesn't mention Physalis pruinosa!) makes us think that CYP87G1, CYP88C7, and CYP749B2 are from Withania somnifera – but would the Withania somnifera genes have the same names? How similar are they? Based on the tree and the naming it is not even clear which are the orthologs of the ones being tested here.

Response: Thank you for your comments. We improved the narrative by focussing more on *W. somnifera*, for example by performing VIGS in this plant, and by confirming the functional conservation of the three *CYP*s and *SDR*s within and between species. Homologues from both sub gene clusters of *W. somnifera* were tested and shown to have the same function as the previously identified genes from *P. pruinosa* (**Supplementary Fig. 19, Fig. 7d, Supplementary Fig. 24**).

In the same light, we also improved the nomenclature of *CYP*s. The *CYP* names were assigned by Dr David Nelson. The *CYP*s that we showed to be functionally equivalent do indeed bear the same name, because they are extremely similar and belong to the same orthogroup. Wherever necessary, we added a species abbreviation to indicate the biological origin (e.g., *WsCYP87G1*) for clarity. This is now explained in the manuscript:

New version: As all of these functionally equivalent *CYP*s belong to the same orthogroup, they were assigned the same systematic name *CYP87G1* and

were further distinguished by species codes (*Ws* or *Pp*) and internal names (see Supporting File).

There are some CYPs in the gene cluster that belong to a different orthogroup than CYP88C7 (e.g., CYP88C8); our data shows that these are not functionally equivalent to CYP88C7 (**Supplementary Fig. 19**). We also added a protein sequence identity matrix (**Supplementary Fig. 18**) for all CYPs which are part of the withanolide gene clusters in the species shown in Fig. 3. To better distinguish between multiple gene copies in the gene clusters, we use an internal naming system that was used in the phylogenetic tree originally shown in Supplementary Fig. 6. Please kindly note that we also provide a Supplementary File that includes all genes from the gene clusters compared by us with their names and functional link. To facilitate the interpretation of supplementary material, we added the names used in the main manuscript to the phylogenetic tree (now **Supplementary Fig. 4**).

*There are other groups working on the same cluster (<https://www.biorxiv.org/content/10.1101/2024.10.02.615186v2>). They focus on the evolution and regulation of the cluster. To distinguish the papers, and to aid in writing a more focused manuscript, perhaps the work here could be refocused on *Physalis pruinosa* biochemistry plus the synthetic biology. The *Withania* genome and phylogenomics could be less of a focus. If the authors can convincingly show withanolide lactone synthase then this would elevate the paper further and make it suitable for publication here.*

Response: As described above, we strengthened the biochemical aspects of our work and focused more on this part of our work. We also strengthened the characterisation of genes from *W. somnifera* to highlight the relevance of our genome sequence. The phylogenomics part was slightly toned down as suggested by the reviewer. As described above, the mechanism of lactone formation was also studied in detail.

General critique is a lack of information in figures and figure legends. The plots need to be interpretable by non-experts and provide sufficient information.

Response: We carefully checked our figures and figure legends again. Further details were added on multiple instances to make them easier to understand (e.g., Fig. 3, Fig. 6). We also extended the figure legends in the supplementary material by explaining all figure elements in detail.

Fig 3 need contig/chrom numbers and locations of the genes/clusters. Perhaps a number showing start and end of syntenic region. Is it a species tree on the LHS?

Response: We added a modified version of the synteny plot with real gene distances as **Supplementary Fig. 2** that contains the contig/chromosome ID, the start position, and the end position of the syntenic region.

The phylogenetic relationship on the left-hand side in this figure is based on a species tree derived from DOI 10.1016/j.xplc.2023.100595. This is now explained in the figure legend of Fig. 3.

Generally for LCMS plots – what is the y axis and for TIC/EIC how is it normalized? E.g. Fig4b, 5bc, 6c. The Relative levels of 6b are a little confusing too – yeast and benth are plotted together but they are normalized separately? This needs a little more clarification in the legend. Could we get some information about the peak sizes overall? Even without standards, knowing whether we are at 10³ or 10⁶ is quite informative.

Response: All GC-MS samples included 5 α -cholestane as an internal standard for normalisation; further normalisation was carried out based on dry weight (either yeast dry cell weight or *N. benthamiana* leaf dry weight). Our LC-MS data were not used for quantitative analyses, but we now specify the maximum ion intensities per LC-MS chromatogram in the newly added **Fig. 7** to provide a better insight into the metabolite levels as suggested.

The data shown previously in Fig. 6b is indeed normalized separately. As suggested, we revised the figure legend of Fig. 6 to describe more clearly how we analyzed the data. Of course, no direct quantitative comparison between metabolite levels in yeast and *N. benthamiana* can be deduced from these values. However, we believe that this is the best way to present which metabolites were found in both hosts and which were only found in one of them.

line 152 – what does “phylogenetically related” mean? Closely related would be better or “within the same subfamily” etc.

Response: Thank you for pointing this out. We rephrased this sentence for clarity.

Old version: For comparison, we included genome sequences of *Solanum lycopersicum*, *Solanum tuberosum*, and *Nicotiana tabacum*, which are phylogenetically related, but not known as withanolide producers.

New version: For comparison, we included genome sequences of **non-withanolide producing Solanaceae species, namely *Solanum lycopersicum* and *Solanum tuberosum* from the same subfamily Solanoideae as well as *Nicotiana tabacum*.**

Genomics – supplementary table S1 – can you show full BUSCO (complete, dup, frag, missing) data?

Response: We have added the requested BUSCO details to **Supplementary Table 1**.

Clarify line 124 – did you also reannotate TEs/repeats?

Response: Yes, we performed an annotation of different repeats using EDTA and TRF. A section about the methodology of the TE/repeat annotation was added to the method section.

Condense some text – e.g. paragraph at 178-197

Response: Thank you for the suggestion. We condensed the paragraph and also moved parts of it to the discussion section as suggested by Reviewer 3.

Reviewer 2:

The work by Hakim et al., titled “Phylogenomics and metabolic engineering reveal a conserved gene cluster in Solanaceae plants for withanolide biosynthesis” is an attempt to understand the genetic organisation of genes involved in withanolide biosynthesis in Solanaceae plants. In this effort, authors have performed genome sequencing and assembly of Withania somnifera and compared it with genomes of withanolide producing solanaceae plants. From this analysis, they found a gene cluster and sub-clusters associated with 24ISO, which has been previously shown to form 24-methylene desmosteraol, a possible precursor for withanolides biosynthesis. Further, authors have shortlisted 3 genes of Physalis pruinosa and functionally co-expressed them with 24ISO in yeast and tobacco along to study their role in withanolide formation. Through the analysis of the products formed by the engineered yeast and tobacco, they claim a pathway for withanoside V aglycone formation. The study is interesting and the efforts are worthwhile in the backdrop of the current lack of understanding of withnaolide biosynthetic pathway. Nevertheless, this study lacks focus and has several shortfalls for publication as listed below;

Response: We thank the reviewer for their valuable comments and for acknowledging that our study is interesting and worthwhile.

1. The major concern of this study is lack of in planta characterization of genes within the host plant. The use of only heterologous system is not an authentic way of deciphering the pathway, it needs to be established in host plant also. Since virus-induced gene silencing and overexpression methods are well established in W. somnifera, authors should have adopted these approaches to prove their claims.

Response: We thank the reviewer for the suggestion. As suggested, we performed virus-induced gene silencing to probe the function of our newly discovered and characterised genes *CYP87G1*, *CYP88C7*, *CYP749B2* and *SDR* in planta in *W. somnifera*. *24ISO* was silenced as a positive control. While overexpression in *W. somnifera* is likely very useful for later pathway engineering applications, we decided that this would not be as beneficial as gene silencing to corroborate the biochemical function of the enzymes that we report in our manuscript.

Virus-induced gene silencing of all genes with the exception of *SDR* caused a drastic decrease in withaferin A levels to 4-22% mean values compared to a

PDS-GFP silencing control as determined by LC-MS analysis (**Fig. 7d**). Simultaneously, we used GC-MS analyses to test accumulation of pathway intermediates upon gene silencing. Indeed, silencing of all genes except for *SDR* led to a substantial accumulation of the pathway precursors that we determined by heterologous expression (**Fig. 7e,f**). Notably, we could also detect the shunt products that we observed in *N. benthamiana* in silenced *W. somnifera* plants; this suggests that these shunt reactions are common in Solanaceae plants and also contribute to the structural diversity of withanolides in their natural producer plants. The fact that silencing of *SDR* only had a minor effect on withanolide levels was surprising to us, but is also in agreement with our data from *N. benthamiana* expression; apparently, both *N. benthamiana* and *W. somnifera* contain background enzymes that can functionally complement the *SDR* from the withanolide gene cluster to some extent. Taken together, our VIGS results not only confirm that CYP87G1, CYP88C7, and CYP749B2 are crucial factors for withanolide biosynthesis in *W. somnifera*; they also support the order of biosynthetic steps, shunt reactions, and background *SDR* activity that we determined by heterologous pathway reconstitution in *N. benthamiana*. All VIGS results are shown in the newly added **Fig. 8**.

2. As per the abstract, it sounds like authors have characterized genes from W. somnifera. However, in the results gene sequences of P. pruinosa have been used for expression in yeast and tobacco model. It is not comprehensible given the fact that efforts were focussed genome analysis of W. somnifera, the most well-known plant for withanolides accumulation.

Response: Thank you for the suggestion. To improve the narrative, we now added functional data for all three CYPs from *W. somnifera* as well. For each of them, we tested both homologues from the two sub gene clusters in *W. somnifera* and showed that they have the same biochemical function (except CYP749B2) (**Supplementary Fig. 19**). We also identified a further enzyme (*SDR*) that is involved in lactone formation (see response to Reviewer 1). Again, *SDR* function is conserved within and across species (**Supplementary Fig. 24, Fig. 7d**). Our data now confirms that the functions of the three CYPs and the *SDR* are generally conserved between the sub gene clusters in *W. somnifera* and between *W. somnifera* and *P. pruinosa*.

3. The authors have used 24ISO as an anchor to identify gene clusters, however there are several genes that are known to affect withanolide production in W. somnifera. How do those genes relate to this gene cluster? Do they have clusters of their own? Are these clusters expressed upon treatment to elicitors like methyl jasmonate?.

Response: We checked the genomic locations of additional genes previously reported to affect withanolide production in *W. somnifera* and not already mentioned in Supplementary Table 7. **Supplementary Table 8** is now linking these additional genes to loci in our genome sequence. None of these previously reported genes is located in the two biosynthetic gene clusters that we discovered in *W. somnifera*. We also compared their expression patterns

to genes in the *W. somnifera* gene cluster, which did not reveal any striking similarity to the expression pattern of 24ISO (**Supplementary Table 9**). Additional studies will be necessary to validate the specific functional roles of these genes in relation to the cluster genes.

Response of the withanolide biosynthesis genes to MeJa was already investigated in *Physalis grisea* in a previous study (<https://doi.org/10.1073/pnas.2420164122>). There are no publicly available RNAseq data for *W. somnifera* upon elicitor treatment; hence, we cannot say which genes are induced by elicitors such as methyl jasmonate.

4. Figure 3 can be replaced with supplementary Fig 1. Do authors know the size of these gene clusters?. If so, needs to be elaborated and a scale should be included in the corresponding figures. The appearance of two gene clusters in W. somnifera is quite intriguing and can be discussed more. Was a sub-gene cluster found in D. stramonium?.

Response: Thank you for the suggestions. We replaced **Fig. 3** with Supplementary Fig. 1 as suggested. Details about the size of the gene clusters are now included in a new **Supplementary Fig. 2**.

We agree that the appearance of the second gene cluster in *W. somnifera* is intriguing, but unfortunately more analyses and biochemical data will be required before this can be properly discussed.

Yes, *D. stramonium* also contains two sub gene clusters. We show this information in Fig. 3. This is further supported by analysis of publicly available gene expression data in Supplementary Fig. 10, showing two distinct groups of expression patterns.

5. There has been abundant work done already to improve triterpene and sterol production in yeast. The authors should review the literature about engineering yeast. Yang et al., (<https://doi.org/10.3390/biom11111710>) for instance has developed a yeast strain capable of producing ~200 mg/L of 24-methylene cholesterol, for this very purpose. Building upon this work would have resulted in a more favourable yeast testing platform. The bottle-neck caused by 7RED could be overcome by screening additional DWF enzymes as done by Yang et al (2021). Interestingly the expression of 24ISO resulted in shunt products while these were not seen in the original paper (Knoch et al., 2018), any explanation why?.

Response: We agree that Yang et al. have done excellent work to produce 24-methylenecholesterol in yeast. Nonetheless, as we show in our manuscript, 24-methylenecholesterol (**1**) first has to be isomerised by 24ISO to 24-methylidestosterol (**2**) before further oxidation can take place. The problem in yeast is not so much that the activity of 7RED is too low; in the absence of 24ISO, 24-methylenecholesterol (**1**) is the main product also in our yeast system (strain KMY77, Fig. 4b), which demonstrates that our 7RED generally works okay. The main issue is that 24ISO is faster than 7RED and shows substrate promiscuity, as it can isomerise not only 24-methylenecholesterol (**1**) but also compound **3**. Therefore, it converts a large part of the substrate of 7RED (compound **3**) to the shunt product **6**.

Accordingly, the example by Yang et al. does not offer any solution to the problem that we encountered and described in our work, which only became obvious because we additionally included 24ISO in contrast to the work of Yang et al. To make this problem clearer, we rephrased some sentences, e.g.:

Old version: The occurrence of the non-reduced products ergosta-5,7,24-trien-3 β -ol (**6**) in KMY23 and ergosta-5,7,24(28)-trien-3 β -ol (**3**) in KMY77 therefore indicates a bottleneck caused by limited activity of *P. peruviana* 7RED in our yeast system.

New version: The occurrence of the non-reduced **but isomerised product ergosta-5,7,24-trien-3 β -ol (**6**) in KMY23 and non-reduced product ergosta-5,7,24(28)-trien-3 β -ol (**3**) in KMY77 therefore indicates a misbalance between the very high activity of *P. peruviana* 24ISO and the limited activity of *P. peruviana* 7RED in our yeast system.**

The yeast strain used by Knoch et al. 2018 (named T21) goes back to a publication from 2014 by Sawai et al. (10.1105/tpc.114.130096). In this publication, the profile of their yeast strain T21 that they show in Figure 3C is very similar to that of our strain KMY77 (*7RED erg4 Δ erg5 Δ*). In Knoch et al. 2018, the latest eluting peak shown in Fig. 5A is 24-methylidesmosterol (**2**). As the shunt product identified by us (compound **6**) elutes substantially later than **2**, it is possible that they also observed the shunt product but just did not show that part of the chromatograms.

6. What is most striking is why the authors did not overexpress HMG-CoA reductase, a known rate limiting enzyme of the sterol pathway in yeast. This was later done for N. benthamiana but not for yeast! It is surprising that the overexpression of UPC2-1 did not improve sterol production in yeast. Authors should confirm its overexpression using qPCR and western blotting.

Response: We agree that overexpressing *HMGR* or a truncated version of it is a very common approach to boost terpene production. We therefore originally tried this strategy as well, in combination with overexpression of other mevalonate pathway genes. The resulting strains produced very high levels of squalene, but not substantially increased amounts of 24-methylidesmosterol. This finding is also in agreement with the work of other groups (e.g., <https://doi.org/10.1021/acssynbio.0c00417>; <https://link.springer.com/article/10.1007/s002530051138>) and emphasises that sterol metabolism in yeast underlies further regulatory constraints which cannot be overcome by simple mevalonate pathway overexpression. Notably, a *tHMGR* gene was also not included in the publication mentioned by the reviewer in their comment above (Yang et al.). To keep the metabolic burden as minimal as possible, we therefore decided not to include overexpression of *tHMGR* or other mevalonate pathway genes in our final strains.

Regarding UPC2-1, we originally chose to include this based on the report by Xiu et al. (<https://doi.org/10.1016/j.biortech.2022.127572>), who reported ca. 40% increased 7-dehydrocholesterol levels. However, there are several

contradicting studies, which – like in our work – did not observe substantial improvements by *upc2-1* overexpression:

- 1) In the study by Gao et al., UPC2-1 did not show a clear effect on triterpenoid production (<https://doi.org/10.3389/fbioe.2022.805429>)
- 2) Hu et al. did not observe any improvement of diterpene levels in yeast with UPC2-1 (<https://doi.org/10.1016/j.ymben.2020.03.011>)
- 3) Likewise, Ro et al. only observed a modest effect of UPC2-1 on sesquiterpene levels (<https://www.nature.com/articles/nature04640>)

We therefore conclude that our results are indeed not contradictory to the state of the art considering the ambivalent data for UPC2-1 in the literature, even though the results are not what we hoped to achieve. More research about UPC2-1 will be necessary to understand in which situations it can be beneficial.

7. As the authors themselves have stated, the transient overexpression of 15 gene using mixed cultures of Agrobacterium is questionable as the level of induction achieved is not always the same. Generation of transgenic tobacco (which has well established transformation protocol) with enhanced substrate production would have been a more robust approach.

Response: Thank you for the suggestion. To the best of our knowledge, no one has yet achieved generation of transgenic tobacco with such a large number of targeted genetic modifications. Besides the practical limitations of this approach, we anticipate that undesired side effects on phytosterol and brassinosteroid levels would lead to severe negative phenotypes and would likely substantially affect the viability of such transgenic tobacco plants. We also strongly believe that our approach is not questionable but indeed state-of-the-art and used by many key studies in the field (e.g., <https://www.nature.com/articles/s41467-023-42253-y>, <https://www.sciencedirect.com/science/article/pii/S1674205223003301>, <https://www.science.org/doi/10.1126/science.adf1017>).

In summary, the authors have heavily relied on the findings of Knoch et al (2018) for this work. Though there is evidence to suggest other phytosterols as the precursor of withanolides, the authors have chosen 24-methylene desmosterol for their genes which have limited their findings. Apart from the genomic resources generated, the engineering of model platforms and characterization of putative genes of the withanolide biosynthetic pathway are still in its nascent stages.

Response: While we cannot fully rule out that other phytosterols might also be precursors for some withanolides, we believe that there is very strong evidence by isotope labelling – independently of the work by Knoch et al (2018) – that 24-methylidestmosterol and 24-methylenecholesterol are indeed the main precursors of withanolides (e.g., <http://xlink.rsc.org/?DOI=c39860001459>, <https://www.sciencedirect.com/science/article/pii/S0031942200843745>). Hence, our focus on 24-methylidestmosterol as a precursor is not a limitation in

our opinion but merely reflects the consensus pathway. Furthermore, this is also supported by our experimental data: Activity of CYP87G1 as the first oxidative step in the pathway was only observed in *N. benthamiana* when the genes leading to production of 24-methyl-desmosterol were included. If another phytosterol such as campesterol, stigmasterol, or β -sitosterol was a suitable precursor, we would have observed activity of CYP87G1 without co-expressing any other gene, because these phytosterols are naturally present in *N. benthamiana*. Likewise, our virus-induced gene silencing data of *24ISO* as a positive control resulted in a strong decrease of withaferin A levels to 22% mean value compared to a *PDS-GFP* silencing control (**Fig. 8d**); this would also not be expected if 24-methyl-desmosterol was not the major key precursor to withanolides.

We would also like to emphasise that our gene discovery work enabled the first successful heterologous production of any withanolide-related compound (here: physalindicanol A, physalindicanol B, phyministerol A, and withanoside V aglycone), three of them even in quantities sufficient for NMR analysis; we believe that this would not have been possible if the model platform engineering and gene characterisation in our manuscript were still in their nascent stages.

Minor points:

The trivial names can be reworked. In their current form, they sound like amino acid substitutions (W22H) rather than catalytic activity. Withanolide 1-hydroxylase is not appropriate as the substrate is not a withanolide.

Response: We understand that the suggested name might be a source for confusion, but from the context it should be clear that these are not amino acid substitutions. Moreover, the same nomenclature has already been used in the general phenylpropanoid pathway (cinnamate 4-hydroxylase, C4H) and in flavonoid biosynthesis (flavanone 3-hydroxylase, F3H), both well studied model systems. We also redefined that the abbreviation “W” stands for “withanolide biosynthesis” rather than just “withanolide” to reflect that these enzymes are part of withanolide biosynthesis.

The supplementary material is too long and complex. The raw spectral data can be shifted to the bottom or to an additional file. Several figures can be clubbed into a single figure.

Response: Thank you for the suggestion. We believe that including all the data is essential to make our research reproducible and comply with open science principles. The raw spectral data are already at the end of the list of figures. We believe that merging several figures into a single one would not reduce their complexity and we would rather leave them separated. However, to provide a better overview over the supplementary data, we now included a **Table of Contents** in this document.

Reviewer 3:

The manuscript by Samuel Edward Hakim et al. employs a phylogenomics approach to identify a conserved gene cluster involved in withanolide biosynthesis and validates the functions of these genes by establishing a genetic engineering system in yeast and tobacco. I read this manuscript with great interest and commend the authors for their extensive efforts on this project. As my expertise lies in genomics rather than biosynthesis, my comments will focus on the genome assembly and phylogenomics sections.

Response: We thank the reviewer very much for their valuable comments and for acknowledging our extensive efforts.

*1. I noticed that a genome of *Withania somnifera* has already been released in NCBI (https://www.ncbi.nlm.nih.gov/datasets/genome/GCA_039654485.1/). A comparison with this NCBI-released genome would be necessary to demonstrate the quality of the genome assembled in this study.*

Response: Thank you for pointing this out. The referenced genome sequence assembly of *Withania somnifera* (GCA_039654485.1) contains over 1.1 million contigs with an N50 of about 70 kbp. This is a highly fragmented assembly. We have included a detailed comparison with our continuous genome sequence comprising less than 100 contigs in **Supplementary Table 2**. Given this difference in continuity of multiple orders of magnitude, we believe that a discussion in the manuscript is not helpful or necessary.

2. There is limited information regarding this genome reported in the results. For example, what is the ploidy level, estimated genome size, and heterozygosity of this genome? Additionally, how many ONT reads were sequenced for de novo assembly, and what is the average read length?

Response: Following this suggestion, we have added some details about the *Withania somnifera* genome sequence in the results. The genome size was estimated by MGSE to be 2.94 Gbp which aligns very well with the assembly size of 2.88 Gbp. A histogram was constructed based on a long-read mapping to the assembly to infer the ploidy which appears to be diploid with low heterozygosity (**Supplementary Fig. 1**). A total of 5.1 million ONT reads was generated with an N50 of 39.4 kb. We have added this information in the manuscript in the beginning of **section “Genome assembly of *Withania somnifera*”**.

New version: To explore the genomic context of withanolide biosynthesis, we generated a genome assembly of *Withania somnifera*, known as a prolific producer of withanolides. The genome size of *W. somnifera* was estimated to be 2.94 Gb. Building on Oxford Nanopore Technologies' sequencing method⁴⁰, a total of 5.1 million reads (N50: 39.4 kb) corresponding to an estimated genome coverage of 34.6x were sequenced. The *de novo* assembly comprised 93 contigs with an N50 length of 71 Mb and a total assembly size of 2.88 Gbp (Supplementary Table 1, Supplementary Table 2). The read coverage depth histogram shows a single peak around the

estimated genome coverage suggesting a diploid genome with low heterozygosity (Supplementary Fig. 1).

3. *Given that the genome was assembled using ONT reads, which typically contain a high level of sequencing errors, the resulting genome may have a significant proportion of errors. This could lead to incorrect gene annotations, including premature stop codons and frameshifts. I suggest that the authors de novo assemble the RNA-seq reads into unigenes using the Trinity de novo package and compare these transcripts with their gene annotations. This would provide a clearer overview of the quality of the gene annotation.*

Response: While ONT raw reads generated on R9 flow cells have an accuracy of around 95-97%, our assembly has a substantially higher accuracy due to all-vs-all read correction in the first assembly phase and following corrections by the assembler. The high completeness of the assembly and annotation are indicated by BUSCO scores of 96.5% complete BUSCO genes. This clearly shows that most gene models are not affected by premature stop codons (if they would exist). In addition, we have identified genes in the biosynthetic gene cluster and do not have any evidence to assume that genes would be missing due to sequencing or annotation errors. A de novo transcriptome assembly based on all available RNA-seq data was generated and compared against our annotation of the *W. somnifera* genome sequence. This de novo transcriptome assembly has a completeness of only 34.0% complete solanales_odb12 BUSCOs and does not contain orthologs for 82% of the genes annotated in our genome sequence. Given this low completeness of the de novo transcriptome assembly, we have decided not to mention this in the manuscript.

As a further support of the practical reliability of our data: For eight genes from our *W. somnifera* genome assembly that we either cloned from cDNA or obtained by gene synthesis we obtained functionally active enzymes upon heterologous expression.

4. *The synteny analysis among different species is quite interesting. The results indicate that no 24ISO orthologue was found in *S. lycopersicum*, *S. tuberosum*, and *N. tabacum*. I am curious about the origin of this gene (24ISO). Did it arise from horizontal gene transfer (HGT), de novo formation from the genome sequence, or was it simply lost in these three species?*

Response: Please note that the evolution of 24ISO was already studied (with a smaller sequence dataset) by Knoch et al. in 2018 (<http://www.pnas.org/content/115/34/E8096>). We have no evidence that horizontal gene transfer played a role for the occurrence of 24ISO homologues. Our phylogenetic analysis re-confirms that 24ISO is a paralog of *DWARF1*, along with two other paralogs, *SSR1* and *SSR2*. 24ISO likely emerged from a gene duplication event from *SSR2* at the basis of the Solanales, but we believe that further detailed studies with an extended dataset will be required to properly date this gene duplication event.

5. In line 162, what do the authors mean by “all conserved genes”? Are these genes within the same syntenic block? How many of these genes are there? Providing this information would help readers better understand the context.

Response: We have rephrased the sentence in the manuscript. Genes covering all conserved functions of the biosynthetic gene cluster were discovered while differences in the arrangement of these genes can be seen in Fig. 3.

Old version: Strikingly, all conserved genes belong to gene families common in plant specialised metabolism, most importantly cytochrome P450 monooxygenases (CYPs), 2-oxoglutarate-dependent dioxygenases (ODDs), short-chain dehydrogenases/reductases (SDRs), and acyltransferases (AT); less expected was the occurrence of sulfotransferase (ST) genes.

New version: Strikingly, all **genes in this syntenic region** belong to gene families common in plant specialised metabolism, most importantly cytochrome P450 monooxygenases (CYPs), 2-oxoglutarate-dependent dioxygenases (ODDs), short-chain dehydrogenases/reductases (SDRs), and acyltransferases (AT).

6. The sentence in lines 168-169, “A few withanolides bearing ...” seems unrelated to the preceding context. Please clarify or revise this section.

Response: Thank you; we rephrased this sentence to improve the connection.

Old version: A few withanolides bearing 3-O-sulphate groups are known, however.

New version: A few withanolides bearing 3-O-sulphate groups are known, however, **and their biosynthesis might involve a dedicated sulfotransferase.**

7. Lines 178-186 read more like a discussion. I recommend moving this portion to the discussion section.

Response: Thank you for the suggestion; we condensed this part as also suggested by Reviewer 1 and moved some parts of it to the discussion section.

8. In lines 199-204, the authors need to provide more details regarding gene expression. How many genes and what samples/tissues were examined? The authors mention observing two clearly separated groups of expression patterns but do not provide details about these groups. How many genes are in each group, and what are their characteristics? Are they highly expressed in some tissues while low in others? Additionally, I strongly suggest that the authors sequence some Hi-C reads, as this could assist in genome scaffolding and help investigate the 3D structure of the genome. This may provide insights into why some of these genes co-express, possibly indicating they are in the same chromatin loop.

Response: We show details about the samples used for gene expression analysis in Supplementary Fig. 9-11. We did observe two sub gene clusters

that show differences in their chromatin structure, but refrained from investigating this in more details as colleagues in Munich are working on this aspect of the withanolide biosynthesis gene cluster (<https://www.pnas.org/doi/10.1073/pnas.2420164122>). There is a difference in gene expression between the two sub gene clusters (Supplementary Fig. 9-11).

Since our assembly has already an impressive continuity (N50 of 71 Mbp, 93 contigs), we do not expect a substantial improvement with Hi-C data. The comparison against other genome sequences (Fig. 3) is already possible with our assembly and the entire biosynthetic gene cluster is located on one contig. As our aim was to elucidate the biochemical reactions in withanolide biosynthesis, we considered the assembly quality as fully sufficient for our purposes.

9. In line 298, where it states "only 24ISO," I recommend adding the Latin name for the gene for clarity.

Response: Thank you, we added a short clarification.

Old version: We first transiently expressed only 24ISO in *N. benthamiana*; while this caused changes in the metabolic profile, no 24-methyl-desmosterol (**2**) was detected.

New version: We first transiently expressed only 24ISO, the gene encoding sterol Δ^{24} -isomerase, in *N. benthamiana*; while this caused changes in the metabolic profile, no 24-methyl-desmosterol (**2**) was detected.

Editorial changes:

We would further like to make the following changes to the author list of our manuscript:

- Arne Bülte-meier should be changed to a shared first author.
- Jessica Eikenberg should be added to the author list.
- Nancy Choudhary and Boas Pucker have recently moved to the University of Bonn and would like to include their current affiliation.

All co-authors agree to these changes and have signed the corresponding form.

We hope that our manuscript will now be considered as acceptable for publication. We look forward to your reply.

Sincerely,

Prof. Dr. Jakob Franke

Prof. Dr. Boas Pucker

Point-to-point response

Please find here our detailed point-by-point response to reviewer 2's comments. All substantial changes in our manuscript and SI have been highlighted in yellow.

Reviewer 2:

The revised manuscript, titled “Phylogenomics and Metabolic Engineering Reveal a Conserved Gene Cluster in Solanaceae Plants for Withanolide Biosynthesis,” represents a significant advancement from its previous iteration. I commend the authors for thoughtfully incorporating the critical suggestions provided by me and the other reviewers, meticulously conducting the necessary experiments, and diligently revising their manuscript. A particularly noteworthy contribution is the discovery of the role of SDRs in the biosynthesis of withanoside V aglycone, which substantially enhances the manuscript's impact.

Response: We thank the reviewer for acknowledging our substantial improvements of the original manuscript.

However, the manuscript still appears disjointed, lacking a cohesive narrative that guides the reader seamlessly through the content. To enhance clarity, every mention of CYP should be accompanied by an abbreviation indicating the species referenced—either “Ws” or “Pp.”

Response: Thank you for the suggestion – we now added species abbreviations where appropriate throughout the manuscript.

The engineering of the yeast platform for 24-methylenedesmosterol production could have been approached with greater deliberation. The authors sidestep deeper analysis by stating that “sterol metabolism in yeast

underlies further regulatory constraints.” However, these constraints are well documented in the literature and have been successfully addressed. For instance, while the authors attempted a simple overexpression of HMGR, which led to a substantial increase in squalene, it failed to enhance 24-methylene desmosterol production. It is well established that sterol accumulation represses downstream genes such as ERG1. Overexpressing these downstream genes could have facilitated higher 24-methylene desmosterol yields.

Response: The aim of our work was to use metabolic engineering to generate a yeast platform that produces 24-methyl-desmosterol and can therefore be used to identify withanolide biosynthetic genes. This aim was successfully completed by us. Of course, future improvements of this yeast system are possible, but outside of the scope of our manuscript.

Besides, we do not think that the constraints underlying sterol metabolism relevant for the production of withanolides in yeast are well documented in the literature and successfully addressed. Before our work, 24-methyl-desmosterol has only been produced at analytical scale in yeast (<http://www.pnas.org/content/115/34/E8096>), and no withanolide-related metabolite downstream of it has been produced in yeast at all. Therefore, these points could not have been addressed before. Of course, useful information for future work can be gleaned from metabolic engineering of other steroids in yeast.

Additionally, UPC2.1 serves as a master regulator for multiple genes, including those involved in precursor utilization. The studies cited by the author in their response have employed UPC2.1 overexpression without simultaneously downregulating competing pathway genes. A notable quote from their cited paper states that “the combination of downregulating ERG9 and overexpressing upc2-1 increased amorphadiene production” (<https://doi.org/10.1038/nature04640>). In this study, the competing gene, ERG5 was knocked out, which raises an important question: Why did UPC2.1 overexpression not influence 24-methylene desmosterol production? The authors should provide a more convincing explanation for this observation.

Response: Our aim was to generate a yeast platform that would allow us to identify withanolide biosynthetic genes, and this was successfully achieved. Combining downregulation of ERG9 with UPC2.1 overexpression is no option in our case, because Erg9p (squalene synthase) is crucial for the production of steroids. Erg5p (C-22 sterol desaturase) is biochemically not related to Erg9p, and hence, conclusions from the combination of ERG9 downregulation and UPC2.1 overexpression cannot be transferred to our system. Studying why UPC2.1 overexpression did not influence 24-methyl-desmosterol production would require a lot of additional work that is far outside of the scope of this work – the elucidation of withanolide biosynthesis.

I strongly encourage the authors to inform readers about the mevalonate pathway engineering in yeast that failed to increase 24-methylene desmosterol production, as this would help prevent unnecessary duplication in future research efforts.

Response: Thank you for the suggestion. We now added this information to our manuscript.

New version: In initial experiments, we observed that overexpression of mevalonate pathway genes only resulted in increased levels of squalene but not of downstream sterols. Therefore, to keep the metabolic burden on our strains as low as possible, no mevalonate pathway genes were overexpressed.

*Figures 6a and 7a could be combined into a separate figure at the end, illustrating a well-constructed (proposed) pathway from 24-methylene desmosterol to withanoside V aglycone, including its shunt products, in yeast and *N. benthamiana*. This would unify the study and effectively convey the role of each CYP and SDR in the pathway.*

Response: We discussed the option to combine Fig. 6a and 7a. However, we believe that this would not simplify or improve our manuscript. Either, we would show the structures only a single time in a merged pathway, and then some of the figures would not include the corresponding structures anymore, which would make them more difficult to understand; or, we would show the structures multiple times, but this would make our manuscript even longer and more redundant. We therefore decided to keep the pathway separated as is.

*A key question raised in my previous review remains unresolved: Do the enzymes studied by the authors establish the pathway under laboratory conditions, or is this the naturally occurring pathway in plants? This was the primary reason for recommending in planta characterization studies via VIGS and overexpression. The authors have conducted VIGS on the equivalents of the three CYP and SDR genes in *W. somnifera*, but they have co-silenced PDS as a visual marker for gene silencing. Consequently, this only validates the role of the CYP/SDR genes within a PDS-silenced background. Given that PDS silencing is known to have collateral effects on the metabolome, the CYP and SDR genes should be silenced independently of PDS to avoid confounding results.*

Response: The reviewer is correct that *PDS* silencing has an effect on the metabolome, and we had already identified this with extra controls that we had not shown in our revised manuscript before. Of course, all of our analyses in Fig. 8 were therefore already previously based on *PDS-GFP* and not *GFP* alone to account for this effect. We now show these **extra controls** (silencing of *GFP* alone; silencing of *PDS* alone) in **Supplementary Fig. 25**. This data illustrates that the effect of *PDS* silencing alone on withaferin A levels (ca. 45% decrease compared to the nonspecific *GFP* control) is clearly weaker than what we observe when we additionally silence *24ISO*, *CYP87G1*, *CYP88C7*, or *CYP749B2* (ca. 90-95% decrease compared to the *GFP* control). Notably, co-silencing of *PDS* with either *GFP* (nonspecific control) or *SDR* (found to be redundant in *N. benthamiana*, Fig. 7b/d) did not lead to a further decrease of withaferin A levels compared to *PDS* silencing alone, indicating that the metabolic effects are indeed specific for the true target gene. Comparable results were obtained in two independent batches of VIGS with *PDS* co-silencing.

Furthermore, we strongly believe that this *PDS* co-silencing approach does not represent a confounding factor that would invalidate or weaken the conclusions of our VIGS experiments:

1) **Multiple groups have independently used such a *PDS* co-silencing approach in different plants and different pathways.**

- *Catharanthus roseus*, monoterpene indole alkaloids: <https://academic.oup.com/plphys/article/187/2/846/6306042>
- *Lupinus angustifolius*; quinolizidine alkaloids: <https://link.springer.com/article/10.1186/s13007-021-00832-4>
- *Antirrhinum majus*; developmental genes <https://plantmethods.biomedcentral.com/articles/10.1186/s13007-020-00683-5>

In all of these publications, *PDS* co-silencing was considered as a positive way to improve the sensitivity and statistical reproducibility of VIGS. The **main requirement is that the metabolic effect on the target metabolites caused by *PDS* silencing is smaller than the metabolic effect caused by silencing of the true target genes** (<https://link.springer.com/article/10.1186/s13007-021-00832-4>) – **which is supported by our data** (Supplementary Fig. 25). *PDS* co-silencing can therefore be considered as state-of-the-art and suitable in our system.

2) VIGS in *W. somnifera* typically leads to **patchy and incomplete silencing phenotypes** as judged by the distribution of photobleached areas in *PDS*-silenced plants. We have previously used VIGS without *PDS* co-silencing as suggested by the reviewer, but the silencing effects were much weaker and much less reliable (<https://repo.uni-hannover.de/items/e9723189-2be9-42ac-abd5-6c80c7fed2b4>).

During the optimisation of our *PDS* co-silencing experiments, we also **harvested non-photobleached leaf parts to determine if they also exhibited reduced withanolide levels**. This **preliminary data** showed that there was a clear and strong correlation between the photobleaching phenotype and reduction of withaferin A levels:

For the final data presented in Fig. 8, we therefore used the **photobleaching phenotype from *PDS* co-silencing as a visual guide for sample selection** to identify leaf parts where silencing was successful. **Without *PDS* co-silencing, we would not have such a visual guide for sample selection**; samples would therefore be randomly taken from silenced as well as non-silenced leaf areas, which would lead to highly variable results. Alternatively, we could pool multiple leaves to mix silenced and non-silenced regions, but this would dilute the silencing effects that we can observe.

Repeating VIGS without *PDS* co-silencing as requested by the reviewer would therefore require us to use many more replicates to achieve sufficient statistical power and would likely lead to much less pronounced metabolic effects. In our opinion, this would not strengthen our findings in any way.

- 3) We also monitored the accumulation of intermediates in silenced plants in comparison to a *PDS-GFP* silenced background (Fig. 8e/f). While *PDS* silencing alone does lead to a decrease of withaferin A levels (Supplementary Fig. 25), this **collateral effect does not lead to a substantial accumulation of any withanolide-related pathway intermediate** (Fig. 8e/f, top chromatograms). In contrast, **when we co-silence target genes from the gene cluster, there is strong accumulation of intermediates in excellent agreement with our biosynthetic model**. This data unequivocally confirms that – despite the collateral effect of *PDS* co-silencing on withaferin A levels overall – this co-silencing approach still enables us to observe clear metabolic effects that are solely dependent on the true target gene. Thus, *PDS* co-silencing clearly is no confounding factor.

Considering these arguments, we strongly believe that our VIGS data is fully reliable and hence provides the **evidence requested by the reviewer that all three *CYP* genes are central for withanolide biosynthesis in *W. somnifera*** (Fig. 8d, Supplementary Fig. 26). The absence of a clear metabolic effect during *SDR* silencing (Fig. 8d) is also in agreement with the observation that the *SDR* in *N. benthamiana* is not essential (Fig. 7b). Finally, the fact that we observed accumulation of the same intermediates that we already found in *N. benthamiana* and in yeast and not any other compounds (Fig. 8e/f) provides excellent support for the order of biosynthetic transformations deduced by us in the heterologous hosts. Our **VIGS data therefore does indeed strongly support that this is the naturally occurring pathway in plants**.

The authors have exclusively quantified withaferin A levels in the silenced plants, but this raises an important question—are they suggesting that withanoside V aglycone acts as a precursor to withaferin A? Clarification on this point is necessary.

Response: Silencing of the genes required for the formation of withanoside V aglycone leads to a strong decrease of withaferin A levels in *W. somnifera*, so there is clearly a correlation. However, we do not yet have any direct evidence that withanoside V aglycone is a precursor to withaferin A. This would require knowledge of the genes and enzymes downstream of withanoside V aglycone that possibly connect withanoside V aglycone with withaferin A. As these genes are not known so far, it is not possible to clarify this point in the scope of this study.

A more comprehensive investigation into the fate of other withanolides and withanosides in the silenced plants would greatly enhance the study's depth. Given that many withanolides co-elute with withaferin A in the same LC chromatogram, their quantification should be straightforward. The authors must extend their analysis to include these additional withanolides and provide a thorough discussion of their findings to shed light on the underlying metabolic dynamics.

Response: We extended our analysis and show that not only withaferin A but also the two other main withanolides present in our extracts (classified as withanolides based on their high-resolution masses and retention times) are affected by CYP gene silencing. We now show this data in a **new Supplementary Fig. 26**. We updated the text to refer to this additional data:

New version: A similar metabolic effect was observed for two additional major compounds that were also classified as withanolides based on their high-resolution masses and retention times (Supplementary Fig. 26).

While our data now confirms that CYP gene silencing not only affects the formation of withaferin A but also of two other withanolides, we believe that further discussing metabolic dynamics at this stage is not possible. This would require in-depth knowledge of all subsequent biosynthetic genes and detailed metabolomics studies that are outside of the scope of this work.

The legend in Figure 8 should explicitly mention the names of compounds 1–15 for clarity.

Response: Thank you for the suggestion. We added the names of the compounds as suggested.

Line 517: Sufficient acknowledgement to reviewer 1 should be given here.

Response: We acknowledged reviewer 1 in the acknowledgements section. According to their review of our revised manuscript, the person is pleased with this variant.